https://doi.org/10.1038/s42003-023-04740-2　　OPEN
# Tonian carbonaceous compressions indicate that *Horodyskia* is one of the oldest multicellular and coenocytic macro-organisms

Guangjin Li[1], Lei Chen [2✉], Ke Pang [1,3,4✉], Qing Tang[5,6], Chengxi Wu[1,3], Xunlai Yuan[1,3], Chuanming Zhou[1,3,4] & Shuhai Xiao [7]

Macrofossils with unambiguous biogenic origin and predating the one-billion-year-old multicellular fossils *Bangiomorpha* and *Proterocladus* interpreted as crown-group eukaryotes are quite rare. *Horodyskia* is one of these few macrofossils, and it extends from the early Mesoproterozoic Era to the terminal Ediacaran Period. The biological interpretation of this enigmatic fossil, however, has been a matter of controversy since its discovery in 1982, largely because there was no evidence for the preservation of organic walls. Here we report new carbonaceous compressions of *Horodyskia* from the Tonian successions (~950–720 Ma) in North China. The macrofossils herein with bona fide organic walls reinforce the biogenicity of *Horodyskia*. Aided by the new material, we reconstruct *Horodyskia* as a colonial organism composed of a chain of organic-walled vesicles that likely represent multinucleated (coenocytic) cells of early eukaryotes. Two species of *Horodyskia* are differentiated on the basis of vesicle sizes, and their co-existence in the Tonian assemblage provides a link between the Mesoproterozoic (*H. moniliformis*) and the Ediacaran (*H. minor*) species. Our study thus provides evidence that eukaryotes have acquired macroscopic size through the combination of coenocytism and colonial multicellularity at least ~1.48 Ga, and highlights an exceptionally long range and morphological stasis of this Proterozoic macrofossils.

[1] State Key Laboratory of Palaeobiology and Stratigraphy, Nanjing Institute of Geology and Palaeontology and Center for Excellence in Life and Palaeoenvironment, Chinese Academy of Sciences, Nanjing 210008, China. [2] College of Earth Science and Engineering, Shandong University of Science and Technology, Qingdao 266590, China. [3] University of Chinese Academy of Sciences, Beijing 100049, China. [4] University of Chinese Academy of Sciences, Nanjing 211135, China. [5] State Key Laboratory for Mineral Deposits Research, School of Earth Sciences and Engineering, Nanjing University, Nanjing 210023, China. [6] Department of Earth Sciences, The University of Hong Kong, Pokfulam Road, Hong Kong, China. [7] Department of Geosciences and Global Change Center, Virginia Tech, Blacksburg, VA 24061, USA. ✉email: leichen@nigpas.ac.cn; kepang@nigpas.ac.cn

Proterozoic macroscopic fossils are crucial for our understanding of the early evolution of eukaryotes, particularly in terms of how and when early eukaryotes developed complex multicellular grades of organization and acquired body sizes that are visible to the naked eyes[1–3]. However, fossils at the millimetric-centimetric scale are relatively scarce in successions older than the Neoproterozoic Era and their biogenicity and phylogenetic positions often remain contentious, hampering our ability to reconstruct the tempos and modes of early eukaryote evolution[4]. *Horodyskia* represents one of few examples with a fossil record extending from the early Mesoproterozoic Era (~1.48 Ga)[5,6] to the terminal Ediacaran Period (~550 Ma)[7–12]. This genus is characterized by a string of beads with uniform size and spacing, and was first discovered as "enigmatic bedding-plane markings" from the ~1.48 Ga Appekunny Formation, Belt Supergroup in Montana, United States[5,6]. Subsequently reported occurrences extend the stratigraphic range of *Horodyskia* and *Horodyskia*-like fossils (e.g., *Parahorodyskia* and *Longbizuiella*) from the Mesoproterozoic Era to the Ediacaran Period. These occurrences came from the 1.42–1.26 Ga Balfour Subgroup of the Rocky Cape Group in Tasmania[13,14], the 1.17–1.07 Ga Backdoor and Stag Arrow formations of the Bangemall Supergroup in Western Australia[15–17], the 550–539 Ma Zhengmuguan, Liuchapo, and Piyuancun formations in China[7–12], and possibly the late Ediacaran Kauriyala Formation, Krol Group in India[18]. However, there had been no reports of the occurrence of *Horodyskia* from the Tonian Period (~1000–720 Ma).

*Horodyskia* has a wide global distribution and long stratigraphic range, but its biological affinity remains controversial. Various opinions concerning about the nature of *Horodyskia* have been proposed, including dubiofossil[5] or pseudofossil[19–21], "enigmatic structure"[5], brown alga[15], *Armstrongia*-like sponge[22], colonial eukaryote[23], archaeal–bacterial consortium[2], agglutinated foraminifer[8,12], *Geosiphon*-like fungus[24], testate amoeba[9], and green alga[12]. One of the prominent features of *Horodyskia* in previously reported specimens is the cast-and-mold preservation in fine-grained siliciclastic rocks or in cherts (silicification), with no evidence for the preservation of organic walls[8,13,23,25]. This feature further impedes the biological and phylogenetic interpretation of *Horodyskia*.

Here we report freshly excavated *Horodyskia* specimens from the Tonian Shiwangzhuang Formation (~850–720 Ma) in western Shandong and Jiuliqiao Formation (~950–720 Ma) in Huainan region, North China (Fig. 1; see Supplementary Note 1 for description of stratigraphic background). The new specimens are unique in their diverse preservational styles, including carbonaceous compressions, three-dimensionally preserved organic-walled fossils, shallow impressions, and casts and molds. They were observed and characterized using light microscopy (LM), scanning electron microscope (SEM) with both backscattered electron (BSE) and secondary electron (SE) detectors, energy dispersive X-ray spectroscopy (EDS) elemental mapping, and Raman spectroscopy (see Methods). The data provide a bridge, both chronologically and morphologically, between Mesoproterozoic *Horodyskia moniliformis* specimens and Ediacaran *Horodyskia minor* specimens. The preservation of organic walls in the Tonian *Horodyskia* fossils also reinforces their biogenicity and the Tonian specimens herein aid the phylogenetic interpretation of *Horodyskia* as a colonial giant-celled protist and probably a coenocytic alga. Together with other reports of Mesoproterozoic and Ediacaran materials, this study highlights an exceptionally long range and morphological stasis of this Proterozoic macrofossil taxon and indicates that a coenocytic and colonial body plan dates to at least as old as 1.48 Ga.

## Results

Specimens of *Horodyskia* are unbranched and uniseriate chains consisting of beads (usually preserved as compressed organic-walled vesicles) with relatively constant size and spacing (Figs. 2–5). The chains can be straight, curved, or sinuous. The beads are sometimes surrounded by a halo of lighter-colored material. Based on bead diameter, the *Horodyskia* specimens are described under two species, *H. moniliformis* (present in the Shiwangzhuang Formation; Figs. 2a, c–f, h, l; 3a, d–f, i–p; 4a, g) and *H. minor* (present in the Shiwangzhuang and Jiuliqiao formations; Figs. 2a, b, f; 5), with a cutoff bead diameter of 0.8 mm (Fig. 6; Supplementary Data 1). Average bead diameter of individual *Horodyskia* string shows a positive correlation with both average spacing (i.e., the gap between adjacent bead boundaries; Fig. 6b) and average inter-bead distance (i.e., distance between centroids of adjacent beads; Fig. 6c). A positive correlation between average bead diameter and average inter-bead distance also applies to *Horodyskia* from other localities (Fig. 6c).

*Horodyskia* beads from the Shiwangzhuang and Jiuliqiao formations are preserved on the top bedding surface, typically as carbonaceous compressions (Figs. 2j, k; 4i–k; 5h–l) or as shallow impressions (i.e., negative epireliefs) (Figs. 2g; 3c, d), whereas the halos are preserved as slightly positive epireliefs (Fig. 2g, i, m). Some beads are preserved with slight three-dimensionality (Fig. 2l–p), similar to the three-dimensional preservation of *Beltanelliformis* from the Ediacaran of Central Urals and *Chuaria* from the Ediacaran of South China[26,27], or preserved as casts and molds (Fig. 3o, p), similar to the early Mesoproterozoic Appekunny *Horodyskia* material[23]. Polished slabs and thin sections perpendicular to the bedding surface show that the halos are petrographically indistinguishable from the matrix (Fig. 2k, o, p).

EDS elemental mapping reveals that the halos are enriched in calcium and magnesium, and slightly depleted in aluminum, silicon, and potassium relative to the matrix (Fig. 4a–f; Supplementary Fig. 1a–f), but this compositional pattern is not distinct in vertically polished slabs (Supplementary Fig. 1g–l). The beads are mainly enriched in carbon (Figs. 4h; 5g), consistent with their carbonaceous nature. Raman spectroscopy also shows that carbonaceous material in the beads have D and G peaks that are characteristic of low-grade metamorphism, with D/G peak intensity ratios (roughly equals to apparent intensity ratios at 1350 cm$^{-1}$ and 1600 cm$^{-1}$, i.e., I-1350/1600; *sensu* ref. [28]) of 0.76–0.84 (Fig. 7a; Supplementary Data 2). In addition, principle component analysis (PCA) of the Raman data was used to characterize chemospace distribution of specimens from the Shiwangzhuang Formation. Chemospace distribution of *H. minor* specimens without halos falls within that of *H. moniliformis* specimens without halos (Fig. 7b; Supplementary Data 2). But *H. moniliformis* specimens with and without halos overlap only marginally in their chemospatial distributions (Fig. 7b).

### Systematic paleontology.

#### Genus *Horodyskia* Yochelson and Fedonkin, 2000, emend[6]

2000 *Horodyskia* Yochelson and Fedonkin[6], p. 844.
2022 *Longbizuiella* Yi et al.[11], p. 16, 17.
2022 *Parahorodyskia* Liu and Dong in Liu et al.[12], p. 4, 5.

*Type species*—*Horodyskia moniliformis* Yochelson and Fedonkin, 2000 (ref. [6]).
*Other species*—*Horodyskia minor* Dong et al.[8].
*Emended diagnosis*—Strings of sub-millimeter- to millimeter-sized bead-like (or discoidal structures when compressed) structures. Beads in the same string are uniform in diameter and are uniserially arranged, with a uniform gap in between. A

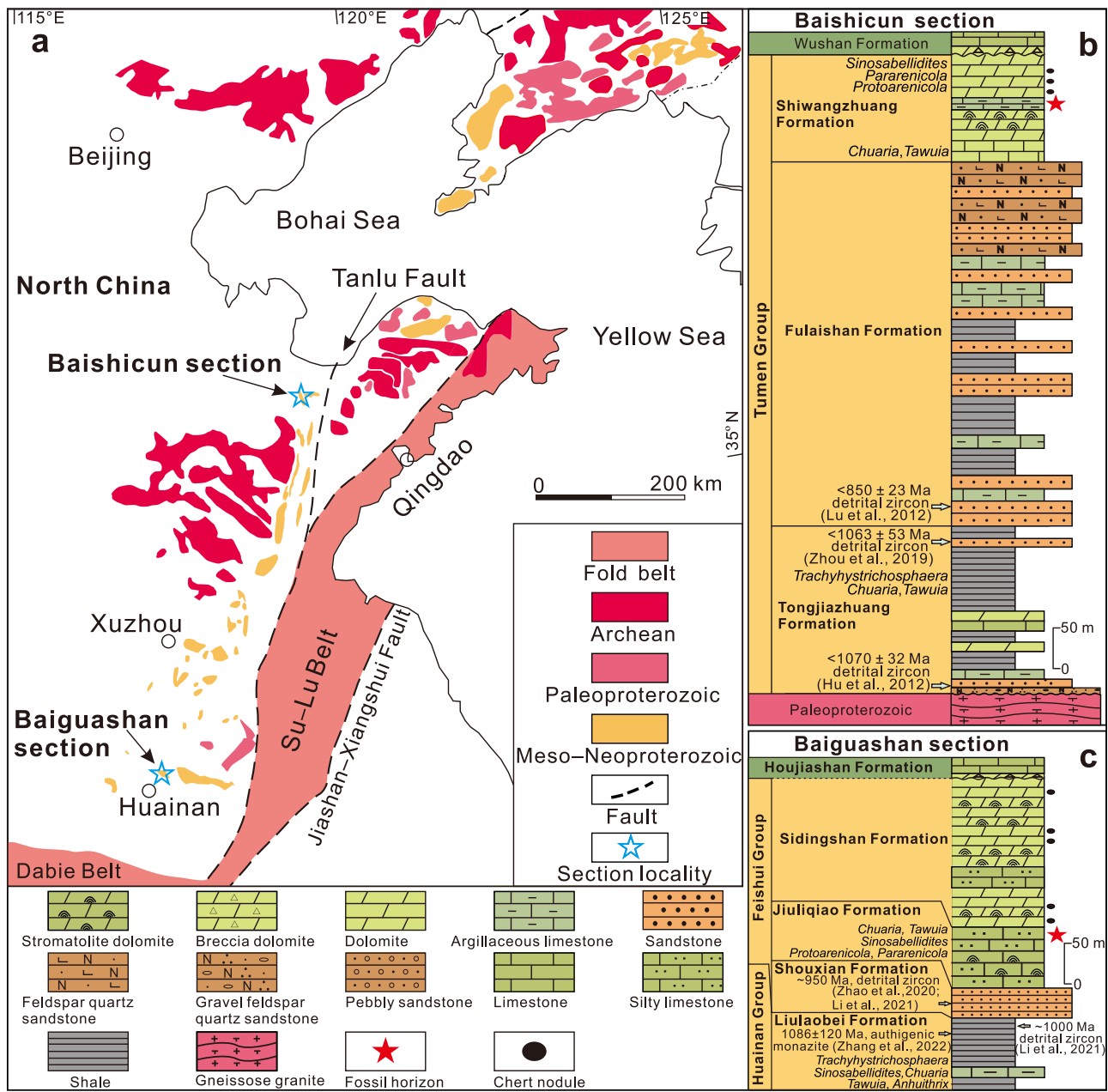

**Fig. 1 Geological map and stratigraphic columns of Baishicun and Baiguashan sections, North China. a** Geological map showing distribution of Precambrian strata in eastern North China Craton, with stars denoting locations of Baishicun and Baiguashan sections. Geological map is modified from refs. [84], [86]. **b** Stratigraphic column of Baishicun section in Anqiu region, western Shandong Province, modified from ref. [83]. Youngest detrital zircon ages from refs. [87–89] provide maximum age constraints. Biostratigraphic data are mainly from ref. [83]. **c** Stratigraphic column of Baiguashan section in Huainan region, northern Anhui Province. Youngest detrital zircon age populations are from refs. [90], [91]; authigenic monazite SIMS Pb-Pb age is from ref. [92]; biostratigraphic data are mainly from ref. [84]. See Supplementary Note 1 for a detailed stratigraphic description.

halo may be present to envelop the beads. A stolon (or filament) may be present to connect adjacent beads.

*Remarks*—Fossils with a string of beads were not treated taxonomically in early studies due to uncertainty about their biogenicity[5],[15]. The genus *Horodyskia* and its type species *H. moniliformis* were established by Yochelson and Fedonkin[6] to account for these fossils, and the type material from the Appekunny Formation in Montana were re-described in greater detail by Fedonkin and Yochelson[23]. The original diagnosis of *Horodyskia* was "presumed colonial organisms of small, vertically oriented, short wide cones, hemispherical on the upper surface, growing from a horizontal tube"[6]. Subsequently, the diagnosis was simplified as "Cones spaced along a horizontal tube"[23].

However, the reconstruction of the beads as "cones" has not been substantiated[19],[24]. The beads were originally spherical, based on observations of *Horodyskia* specimens from the Shiwangzhuang and Jiuliqiao formations and other stratigraphic units[8],[12],[13],[25]. The emended diagnosis recognizes the spherical morphology, as well as the sub-millimetric to millimetric size, of the beads.

A defining feature of the genus *Horodyskia* is the presence of a halo surrounding the beads. This feature has been described from *Horodyskia* specimens preserved in different taphonomic modes, including those preserved in Mesoproterozoic siliciclastic rocks from Montana[23], Western Australia[25], and Tasmania[13], carbonaceous compression specimens from Tonian argillaceous limestone in North China (this paper), and silicified specimens from

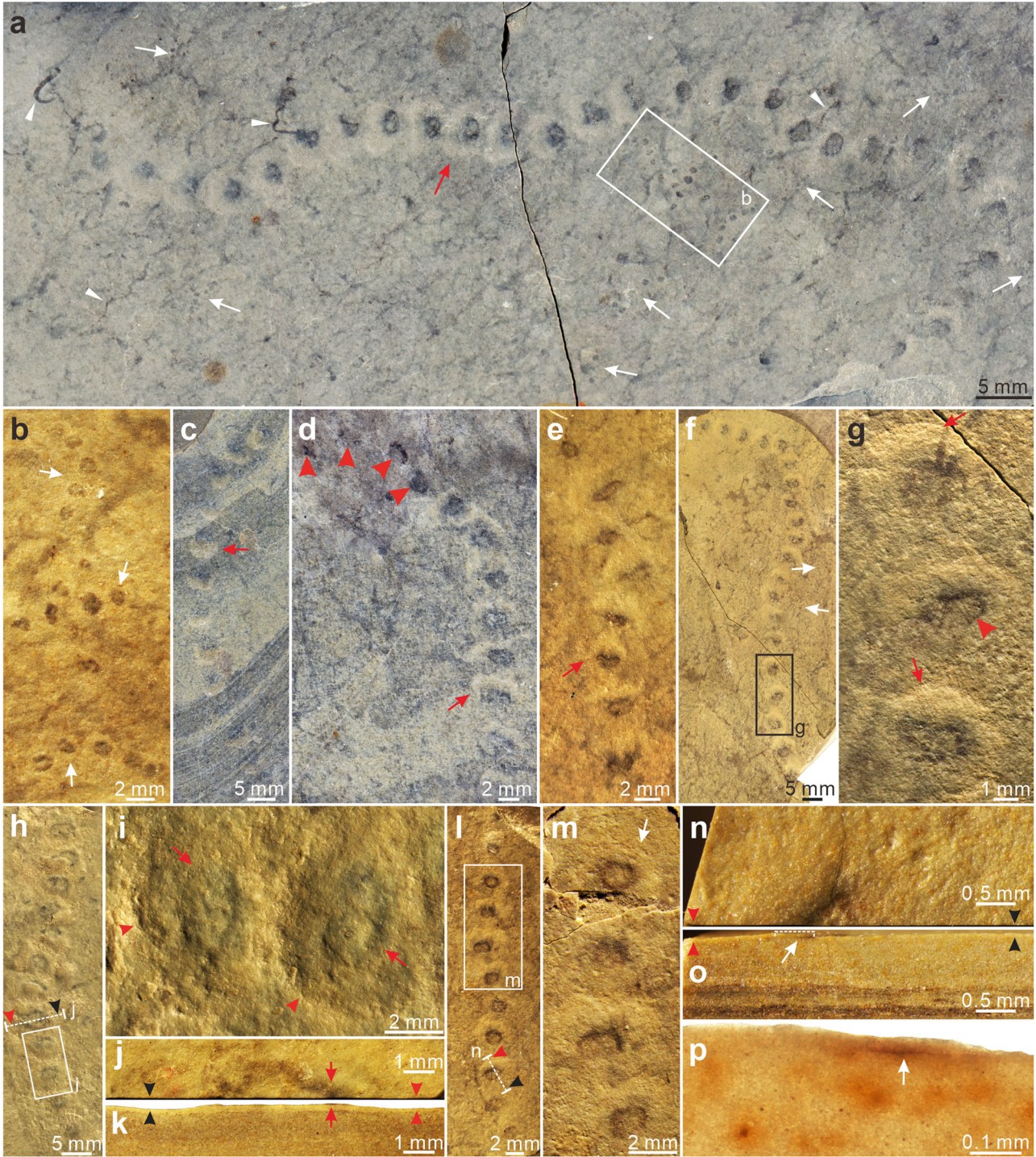

Ediacaran cherts in South China[8,9,11]. The halos have been interpreted as biological structures, such as agglutinated tests[8], gelatinous sheaths[13], or blade-like ribbons of seaweeds[25]. They have also been interpreted as abiotic structures resulting from current activity[15] and organic matter degradation[23].

Considering the consistent occurrence of the halos, we favor a biological interpretation and argue that the halos probably represent fossilized gelatinous matrix surrounding the organic-walled beads and were reinforced by post-mortem mineral precipitation. Contrary to the beads, the halos are not stable structures—they can be present in some specimens but absent in others from the same locality. The positions of the beads are not always located in the center of the halos. Eccentric offset of the bead

centroids relative to the surrounding halos typically occurs toward the same direction along the string (e.g., Figs. 2a, c, d, f; 4a), resulting in the widening and thickening of the halos on the same side[23]. These features indicate that the halos were originally less rigid and more prone to deformation than the beads, and the eccentric offset may be caused by post-mortem shearing by water currents and sediment loading. The lack of a distinct boundary between the halos and the surrounding sediment matrix, as observed in light microscopy of polished slabs and thin sections cut perpendicular to the bedding surface (Fig. 2k, o, p), indicates broadly similar mineralogical compositions. However, there are subtle geochemical differences between the halos and the matrix. For example, in the Shiwangzhuang material of *Horodyskia*, the

**Fig. 2 *Horodyskia moniliformis* and *Horodyskia minor* from the Tonian Shiwangzhuang Formation.** Most beads of *H. moniliformis* are enveloped in a halo. **a** A specimen of *H. moniliformis* with halos (red arrow) preserved together with several specimens of *H. minor* (white arrows) and carbonaceous filaments (arrowheads) on a slab; SWZ-CZ-46. **b** Magnification of box area in **a**; arrows denote three strings of *H. minor*. **c–e** Specimens of *H. moniliformis* with halos (arrows); specimen in **d** possess beads with and without a halo (arrowheads); **c** SWZ-CZ-6; **d** SWZ-CZ-3; **e** SWZ-CZ-44; **f** *H. moniliformis* preserved together with *H. minor* (arrow); SWZ-CZ-53. **g** Magnification of box area in **f**; arrows denote boundary between halo and surrounding matrix, and arrowhead denotes boundary between halo and bead. **h** A specimen of *H. moniliformis* with halos; dotted line labeled "**j**" denotes location where the bead was cut perpendicular to bedding surface; SWZ-CZ-38. **i** Magnification of box area in **h**; arrows denote boundary between halos and beads, and arrowheads denote boundary between halos and surrounding matrix. **j, k** Top view (**j**) and side view (**k**, polished slab) of bead cut along dotted line labeled "**j**" in **h**, with arrowheads denoting boundary between halo and surrounding matrix and matching those in **h**, and arrows denoting bead. **l** A specimen of *H. moniliformis*; dotted line labeled "**n**" denotes location where the bead was cut perpendicular to bedding surface; SWZ-CZ-28. **m** Magnification of box area in **l**; uppermost bead with a halo (arrow) is still discernable after a layer of ca. 0.05-mm-thick sediment was removed, suggesting a degree of three-dimensional preservation. **n–p** Top view (**n**) and side view (**o**, polished slab; **p**, thin section) of bead cut along dotted line labeled "**n**" in **l**; arrowheads in **n**, **o** denote boundary between halo and surrounding matrix and matching those in **l**; arrows in **o**, **p** denote thin carbonaceous film covered by a layer of ca. 0.05-mm-thick sediment. All subfigures are reflected light images on top bedding surface (**a–j**, **l–n**) or perpendicular to bedding surface (**k**, **o**, **p**), except for (**p**), which is a transmitted light image.

halos are enriched in Ca and Mg relative to the matrix (Fig. 4e, f; Supplementary Fig. 1e, f). In other materials, the halos can be enriched in hematite (likely derived from oxidative weathering of pyrite[5,23]) or silica[8,12]. These subtle geochemical differences are probably related to post-mortem mineral precipitation, preferentially within the halos, perhaps induced by the degradation of amorphous gelatinous material in the halos.

Stolons are also present in some *Horodyskia* specimens, best documented in silicified specimens that have been assigned to *Horodyskia minor* or *Parahorodyskia minor*[8,12]. It was also inferred that *H. moniliformis* had stolons that connected adjacent beads[23]. However, structure that connects adjacent beads (i.e., stolon, strand, and filament) have not been observed in the Shiwangzhuang and Jiuliqiao carbonaceous compression specimens, and rarely reported from specimens preserved in siliciclastic rocks[23]. Thus, either the stolons are not preserved in specimens from the Shiwangzhuang and Jiuliqiao formations, or the beads in these specimens were held together by a gelatinous halo rather than a stolon. If the latter case is correct, then the Tonian specimens may be different from the Liuchapo specimens of "*Horodyskia minor*" (or "*Parahorodyskia minor*")[8,12].

Numerous other features have been described in association with *Horodyskia*, including ridges surrounding the beads, linear chevron structures, top depressions or dimple structures, and radiate tubes[5,15,19,23–25]. These have either been considered as uncommon features or attributed to taphonomic variations[13,19,23], and therefore are not included in the emended diagnosis. For example, ridges surrounding the beads are interpreted as an infilled scour mark[23], linear chevron structures are related to current scouring structures[15], top pits or dimple structures are identified as compression features[25], and radiate tubes are considered as an unstable structure since they were not confirmed by subsequent study[19].

We consider *Parahorodyskia* Liu and Dong in Liu et al.[12] and *Longbizuiella* Yi et al.[11] as junior synonyms of *Horodyskia*. *Parahorodyskia* and *Longbizuiella* were erected because of their smaller size and different preservational style from the Mesoproterozoic *Horodyskia*. But the diagnosis of *Horodyskia*[6] does not exclude sub-millimetric beads for *Horodyskia*, and *Horodyskia* specimens with sub-millimetric beads can be preserved together with *Horodyskia* specimens with millimetric beads in the Shiwangzhuang Formation (Fig. 2a, f), suggesting a congeneric relationship. Moreover, the Shiwangzhuang *Horodyskia* material can be variously preserved as carbonaceous compressions (e.g., Figs. 2a; 3a), shallow impressions (Figs. 2g; 3c), three-dimensional organic-walled fossils (Fig. 2l–p), and casts and molds (Fig. 3o, p), indicating that preservational style cannot be used as a criterion to distinguish species.

In this study, we recognize two formal species of *Horodyskia*—*H. moniliformis* and *H. minor*. These two species are mainly differentiated by their average bead diameter, 0.1–0.8 mm in *H. minor* and 0.8–10 mm in *H. moniliformis* (Fig. 6). In specimens from the Shiwangzhuang and Jiuliqiao formations, beads of both *H. minor* and *H. moniliformis* are typically not connected. It is possible that *Horodyskia* specimens with sub-millimetric beads connected by filaments or stolons[8,12] may represent a distinct species, but this taxonomic proposition warrant further investigation.

### *Horodyskia moniliformis* Yochelson and Fedonkin, 2000, emend[6]
Figures 2a, c–p; 3; 4

1982 'String of beads'; Horodyski[5], pl. 1.
1990 'String of beads'; Grey and Williams[15], Figs. 2, 5, 10.
2000 *Horodyskia moniliformis*; Yochelson and Fedonkin[6], p. 844–847, Fig. 1.
2002 'String of beads'; Grey et al.[16], Fig. 2b, c.
2002 *Horodyskia moniliformis* Yochelson and Fedonkin; Fedonkin and Yochelson[23], p. 5–6, Figs. 3, 4, 6–17.
2004 'String of beads'; Martin[29], Fig. 2A, C.
2004 *Horodyskia* sp.; Mathur and Srivastava[18], Fig. 2a.
2010 *Horodyskia williamsii*; Grey et al.[25], p. 14–16, Figs. 2–7.
2010 *Horodyskia williamsii* Grey et al.; Calver et al.[13], p. 23–24, Figs. 5–8.
2019 *Horodyskia moniliformis* Yochelson and Fedonkin; Rule and Pratt[19], Figs. 9–13.

**Referred material.**—Fifty-five specimens (or strings) were examined.

**Locality and horizon**—Argillaceous limestone ~74 m below the top of the Tonian Shiwangzhuang Formation, Baishicun section, Anqiu region, western Shandong Province, North China.

**Emended diagnosis**—A species of *Horodyskia* characterized by relatively large beads, typically 0.8–10 mm in average diameter.

**Description**—Unbranched, uniseriate chains consisting of 3–29 bead-like structures (or discoidal structures when compressed), resembling a string of beads (Figs. 2a, c–f, h, l; 3a, d–f, i–p; 4a, g). Strings can be relatively straight, sinuous, or highly curved, but superimposed and self-superimposed strings are not found. Beads are typically round to oval in shape, but can also be reniform and irregular, with a relatively uniform size. Bead diameter is 0.85–3.53 mm (average = 1.92 mm; SD = 0.55 mm; $n$ = 370 beads from 46 strings; Fig. 6a), whereas average bead diameter in a single string varies from 0.88 mm to 2.99 mm ($N$ = 46 strings; Fig. 6b, c).

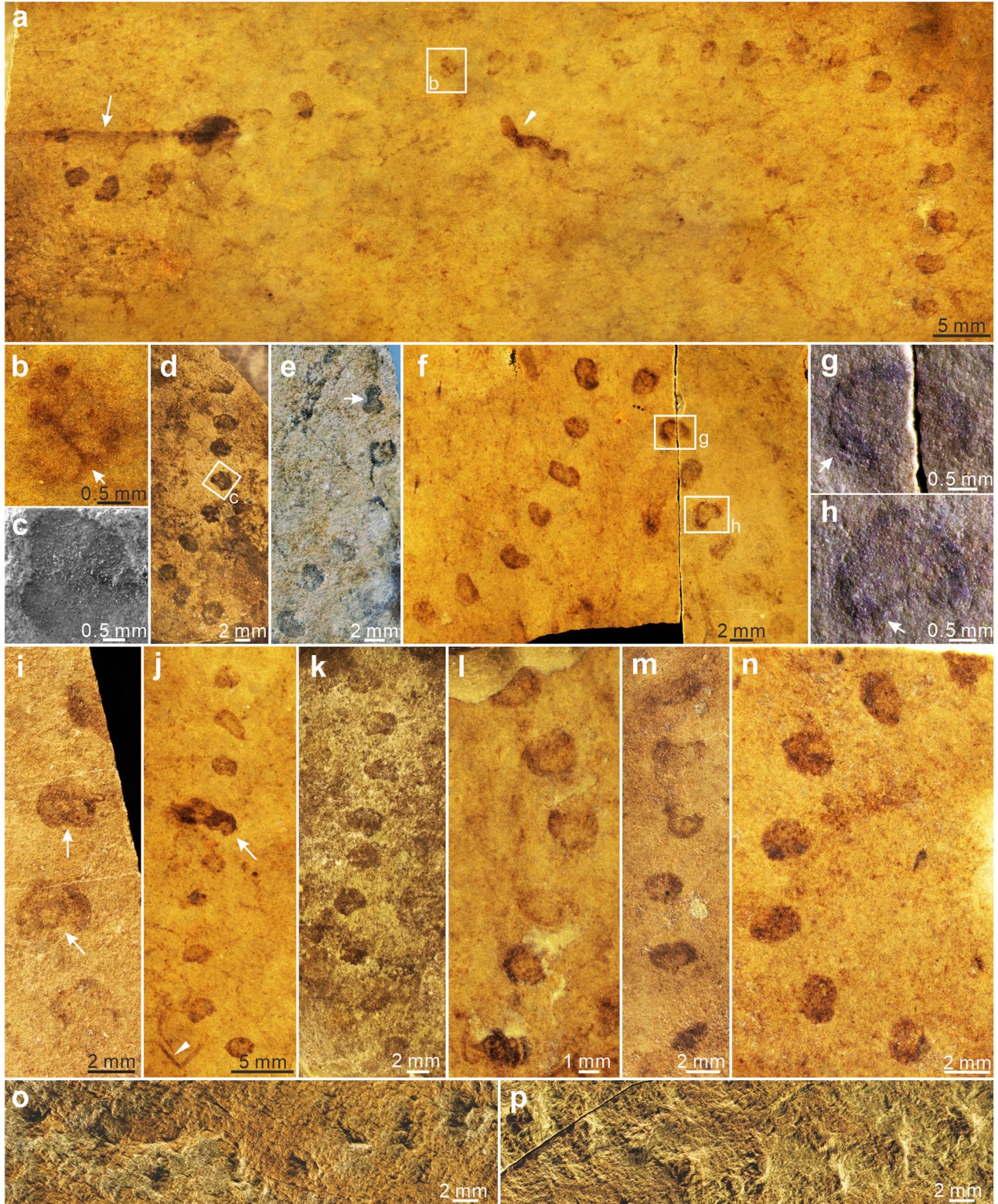

Adjacent beads are separated from each other by a relatively constant space (i.e., the gap between bead boundaries; *sensu* ref. [8]) of 0.82–3.79 mm (average = 2.03 mm; SD = 0.52 mm; n = 130 measurements from 31 strings; Fig. 6b). Inter-bead distance (or distance between the centroids of adjacent beads) is 1.69–9.49 mm (average = 4.03 mm; SD = 0.93 mm; n = 324 measurements from 46 strings; Fig. 6c). In addition, thickenings interpreted as possible

compressional folds (e.g., Fig. 3f–i) and possible binary fission structure (Fig. 3b, e) are also found in the beads.

Beads are sometimes enveloped by a light-colored halo (Fig. 2). When present, the halo usually can be found around all beads in the string. However, there are a few exceptions where beads with and without halos are present in the same string (e.g., Fig. 2d). Halos are circular or oval in shape, and are 3.12–7.41 mm in outer

**Fig. 3 *Horodyskia moniliformis* from the Tonian Shiwangzhuang Formation.** Most beads of *H. moniliformis* are not enveloped in a halo, although a few are surrounded by a faint halo. **a** A long and sinuous specimen of *H. moniliformis* preserved together with a specimen of *Protoarenicola baiguashanensis* (arrowhead) and an unidentified filamentous fossil (arrow); SWZ-CZ-47-1. **b** Magnification of box area in **a**; arrow denotes a bead with a septum, possibly in division. **c–n** Specimens with various bead shapes and varying degrees of string curvature. **c** SEM image of box area in **d**, showing shallow impression of bead preserved as a negative relief. **e** Arrow denotes a dumb-bell shaped bead likely in division. **g, h** Magnifications of box areas in **f**; arrows denote concentric thickening. **i** A specimen consisting of beads with irregular (upper arrow) and concentric (lower arrow) thickening. **j** A specimen of *H. moniliformis* preserved together with filamentous fossils (arrowhead); arrow denotes a bead and a filamentous fossil overlapped with organic film. **d** SWZ-CZ-21; **e** SWZ-CZ-54; **f** SWZ-CZ-40; **i** SWZ-CZ-30; **j** SWZ-CZ-47-2; **k** SWZ-CZ-19; (**l**) SWZ-CZ-43; **m** SWZ-CZ-51; **n** SWZ-CZ-31. **o, p** Part and counterpart; beads are preserved as negative relief on sole bedding surface (**o**) and positive relief on top bedding surface (**p**); SWZ-CZ-58. All subfigures are images on top bedding surface, except otherwise marked, under reflected light microscopy (**a, b, d–p**; RLM) or SEM (**c**).

diameter (average = 4.73 mm; SD = 0.99 mm; $n = 65$ measurements from 9 strings). Generally, halos associated with adjacent beads are in contact or overlap with each other, but there are also a few exceptions where they are separated from each other by a short distance (e.g., Fig. 2c, g).

*Occurrence*—The Tonian Shiwangzhuang Formation, Tumen Group at the Baishicun section, Anqiu region, western Shandong Province, North China; the Mesoproterozoic Appekunny Formation, Ravalli Group, Belt Supergroup, Montana, United States[5]; the Mesoproterozoic Backdoor Formation and Stag Arrow Formation, Bangemall Supergroup, Western Australia[15,16,25]; the Mesoproterozoic Cassiterite Creek Quartzite in the Balfour Subgroup, Rocky Cape Group, Tasmania[13]; possibly the late Ediacaran Kauriyala Formation, Krol Group, Uttaranchal, India[18].

*Remarks*—*Horodyskia williamsii* was established by Grey et al.[25] based on Western Australian specimens that are somewhat different from the type material from the Appekunny Formation, including a different preservational mode, the lack of a growth pattern as inferred for *H. moniliformis* by Fedonkin and Yochelson[23], a decreasing inter-bead distance as beads become larger, and smaller beads with a narrower range of bead size. *H. williamsii* from Western Australian is typically preserved as empty pits on sole surface[25]. However, negative reliefs on the sole surface are also common in *H. moniliformis* from the Appekunny Formation[23] and present in *H. moniliformis* from the Shiwangzhuang Formation (Fig. 3o). Moreover, preservational mode should not be used as a criterion to distinguish species, especially considering that *Horodyskia* can be preserved in different modes and in different lithologies (e.g., silicification in cherts, casts and molds in fine-grained siliciclastic sediments, and carbonaceous compression in argillaceous limestone). The growth pattern established for *H. moniliformis* by Fedonkin and Yochelson[23], i.e., some larger beads keep continuously growing while other intermediate smaller beads cease growing or have regressive development, has not been verified in subsequent studies[19,24]. Therefore, the absence of *H. moniliformis*-style growth pattern in Western Australian specimens, which was regarded as an important feature of *H. williamsii*, is not a solid taxonomic criterion either. The Shiwangzhuang specimens are similar to the Western Australian and Tasmania specimens in relatively small beads with a narrow range of bead size, when compared with *H. moniliformis* from the Appekunny Formation (Fig. 6c). However, bead size of *H. moniliformis* from the Appekunny Formation largely overlaps with that of *Horodyskia* specimens from Western Australian, Tasmania, and North China (Fig. 6c; refs. [13,24]). Therefore, we propose that *H. williamsii* is a junior synonym for *H. moniliformis*, emend the diagnosis of *H. moniliformis* so that it can be objectively differentiated from *H. minor* by its larger bead size (0.8–10 mm in bead diameter), and assign the Shiwangzhuang specimens with bead diameter of 0.8–3.53 mm to *H. moniliformis*.

***Horodyskia minor* Dong et al., 2008, emend**[8]
Figures 2a, b, f; 5

2007 *Horodyskia moniliformis*? Yochelson and Fedonkin; Shen et al.[10], p. 1401–1402, Fig. 4.9–4.12.
2008 *Horodyskia minor*; Dong et al.[8], p. 368–371, Fig. 3a, b, d–l.
2012 *Horodyskia* cf. *minor*; Dong et al.[7], Fig. 3.
2020 *Horodyskia* sp. or *Horodyskia* specimens; Luo and Miao[9], Figs. 3D, F; 5A, B; 6A (but not Figs. 3G; 6B–D; 7F, I).
2022 *Longbizuiella hunanensis* Yi et al.[11], p. 16, 17, Fig. 3.
2022 *Parahorodyskia disjuncta* Liu and Dong in Liu et al.[12], p. 5, 6, Fig. 5A–I (although their figure caption assigned only Fig. 5A–G to this species).
2022 *Parahorodyskia* or taxonomically unidentified specimens, Liu et al.[12], Figs. 11A–D, F; 12A; 13A; 14; 15; 17.
non 2022 *Parahorodyskia minor* (Dong et al.[8]) Liu et al.[12], p. 6–7, Fig. 5J–L.

*Referred material*—More than 100 specimens (or strings) were examined.

*Locality and horizon*—Argillaceous limestone ~74 m below the top of the Tonian Shiwangzhuang Formation, Baishicun section, Anqiu region, western Shandong Province, North China; argillaceous limestone ~14 m below the top of the Tonian Jiuliqiao Formation, Baiguashan section, Huainan region, northern Anhui Province, North China.

*Emended diagnosis*—A species of *Horodyskia* characterized by relatively small beads, typically 0.1–0.8 mm in average diameter.

*Description.*—Specimens from the Shiwangzhuang Formation are preserved on the top bedding surface as carbonaceous compressions. Unbranched string of 3–17 uniseriately arranged bead-like structures (or discoidal structures when compressed) (Figs. 2a, b, f; 5a–l). Strings can be straight or sinuous, and usually occur in aggregation. Constituent beads are round, oval, rod-like, or irregular in shape. The diameter of beads is 0.15–0.66 mm (average = 0.32 mm; SD = 0.10 mm; $n = 218$ beads from 41 strings; Fig. 6a), and average diameter of beads in a single string varies between 0.18 mm and 0.58 mm ($N = 41$ strings; Fig. 6b, c). Inter-bead distance is more or less uniform, 0.26–1.57 mm (average = 0.54 mm; SD = 0.24 mm; $n = 177$ measurements from 41 strings; Fig. 6c), whereas bead spacing (i.e., the gap between adjacent beads; *sensu* ref. [8]) is 0.07–0.89 mm (average = 0.22 mm; SD = 0.12 mm; $n = 126$ measurements from 39 strings; Fig. 6b). Subtle halos enveloping the beads are preserved in some specimens (e.g., Fig. 5b, d), and they are 0.72–1.15 mm in outer diameter (average = 0.90 mm; SD = 0.14 mm; $n = 10$ measurements from 2 strings). Paired beads (e.g., Fig. 5b, d, f) are observed and they are probably derived from binary fission.

Specimens from the Jiuliqiao Formation are also preserved on the bedding surface as carbonaceous compressions. Each string

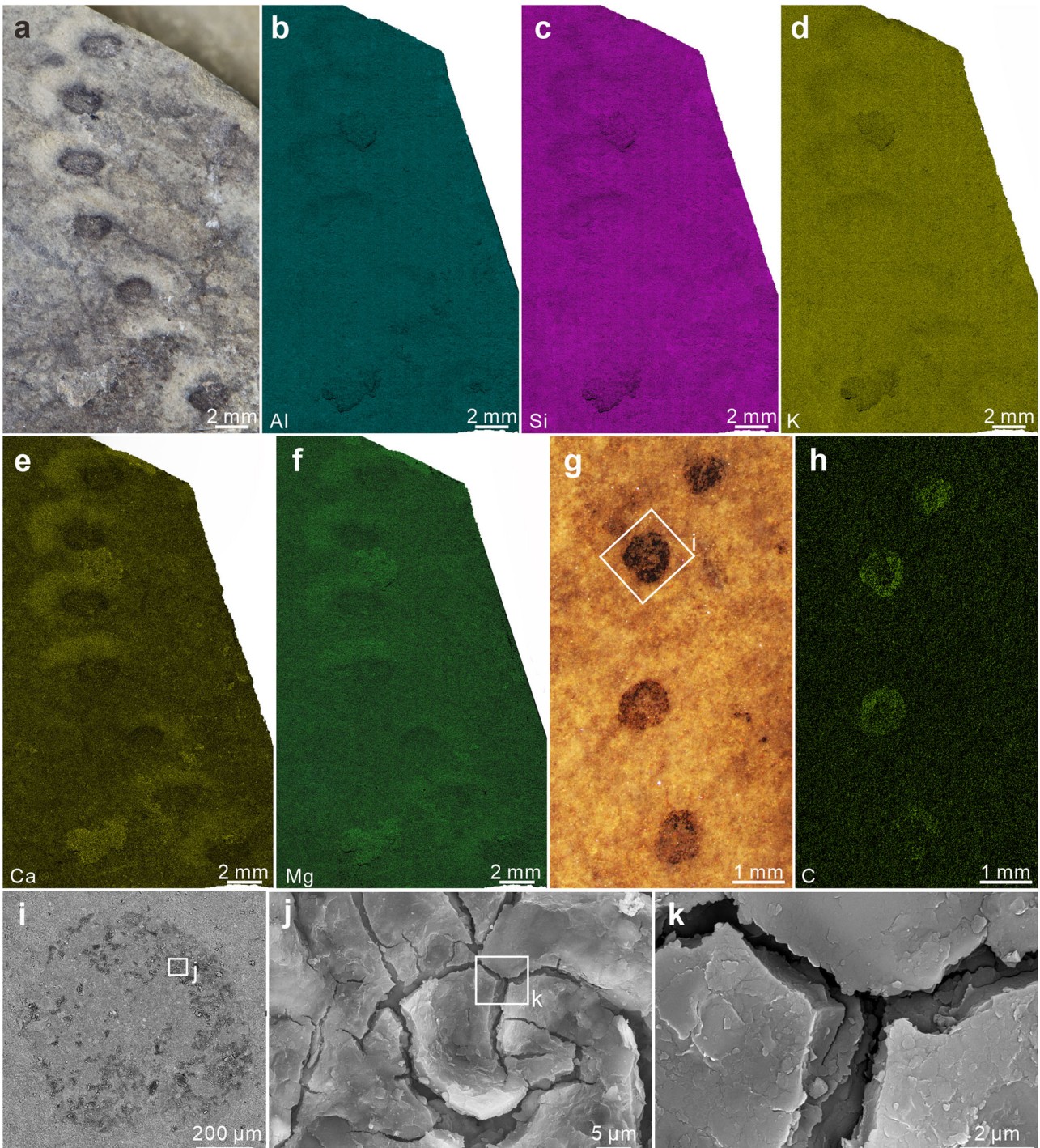

**Fig. 4 RLM images, EDS elemental maps, and SEM images of *Horodyskia moniliformis* from the Tonian Shiwangzhuang Formation. a–f** RLM image (**a**) and EDS elemental maps (**b–f**; element labeled in lower left) of a specimen with halos; halos are enriched in calcium (**e**) and magnesium (**f**), and slightly depleted in aluminum (**b**), silicon (**c**), and potassium (**d**) relative to matrix and beads; SWZ-CZ-25. **g, h** RLM image (**g**), EDS elemental map (**h**; element labeled in lower left), and SEM (**i**, BSE; **j**, **k**, SE) images of a specimen without a preserved halo; beads are enriched in carbon (h); SWZ-CZ-42. **i** BSE image of box area in **g**, showing carbonaceous film of the bead. **j** SE image of box area in **i**, showing multiple cracks in carbonaceous film. **k** Magnification of box area in **j**, showing carbonaceous film which is a few microns in thickness.

consists of 3–11 bead-like structures, and many strings can occur in aggregation (Fig. 5m, n). The beads are round or oval in shape, with their diameter between 0.40 mm and 0.66 mm (average = 0.52 mm; SD = 0.07 mm; $n = 24$ beads from 6 strings), their spacing between 0.13 mm and 0.35 mm (average = 0.24 mm; SD = 0.06 mm; $n = 17$ measurements from 6 strings), and the inter-bead distance between

0.56 mm and 0.98 mm (average = 0.74 mm; SD = 0.10 mm; $n = 18$ measurements from 6 strings). Paired beads and halos are also found in some specimens (Fig. 5m, n). The halos are subtly light-colored and are 0.86–1.63 mm in outer diameter (average = 1.11 mm; SD = 0.19 mm; $n = 27$ measurements from 6 strings). The bead sizes of the Jiuliqiao specimens fall well within the size

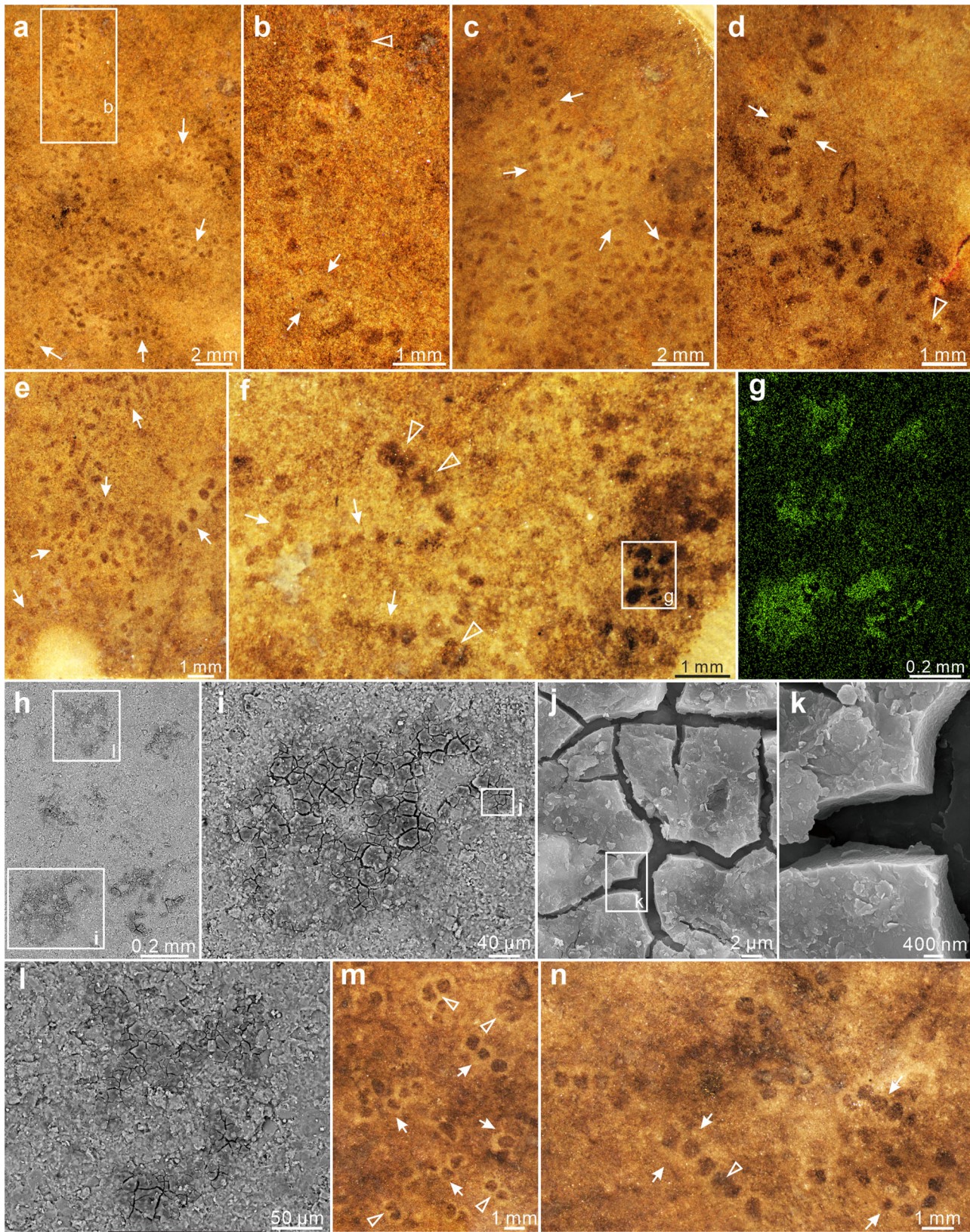

range of the Shiwangzhuang specimens (Fig. 6c). Short strings, each consisting of only two beads, are sometimes found among longer strings (Fig. 5m), but they are similar in the size and shape of beads and halos. In the short strings, beads are between 0.45 mm and 0.66 mm in diameter (average = 0.54 mm; SD = 0.06 mm; n = 18

beads from 9 strings), and light-colored halos are 0.85–1.11 mm in outer diameter (average = 0.97 mm; SD = 0.08 mm; n = 9 measurements from 9 strings).

***Occurrence***—The Tonian Shiwangzhuang Formation, Tumen Group at the Baishicun section, Anqiu region, western Shandong

**Fig. 5 RLM images, EDS elemental map, and SEM images of *Horodyskia minor* from the Tonian Shiwangzhuang and Jiuliqiao formations. a–f** RLM images of specimens (arrows) preserved in aggregations sometimes with halos and paired beads (Triangle) preserved from the Shiwangzhuang Formation. **b** Magnification of box area in **a**, showing specimens with faint halos (arrows) and a bead pair (triangle). **a** SWZ-CZ-23-1. **c** D11-37-11. **d** D10-18-1. **e** SWZ-CZ-23-2. **f** SWZ-CZ-48-2. **g** EDS elemental map of box area in **f**, showing that beads are enriched in carbon. **h** BSE image of the same area in **g**. **i** Magnification of box area in **h**, showing carbonaceous film of the bead. **j** SE image of box area in **i**, showing multiple cracks in carbonaceous film. **k** Magnification of box area in **j**, showing carbonaceous film which is a few microns in thickness. **l** Magnification of box area in **h**, showing carbonaceous film of the bead. **m** RLM images of specimens with faint halos (arrows) and specimens only consisting of two beads (triangles) preserved in aggregation from the Jiuliqiao Formation; BG-J-102-1. **n** Specimen with faint halos (arrows) and a bead pair (triangle); BG-J-102-2. All subfigures are images on top bedding surface.

Province, North China; the Tonian Jiuliqiao Formation, Feishui Group at the Baiguashan section, Huainan region, northern Anhui Province, North China; the late Ediacaran Zhengmuguan Formation, Ningxia Hui Autonomous Region, North China[10]; the late Ediacaran Liuchapo Formation, Guizhou and Hunan provinces, South China[8,9,11,12]; the late Ediacaran Piyuancun Formation, Anhui Province, South China[7].

*Remarks*—The genus placement of *Horodyskia minor* has been debated. *Horodyskia minor* was erected by Dong et al.[8] on the basis of silicified material from the Liuchapo Formation in Guizhou Province of South China, and it is characterized by beads smaller than 0.7 mm in diameter. Silicified specimens of similar size and shape were subsequently reported from the Liuchapo Formation in Hunan Province (as *Horodyskia* sp.[9]) and the equivalent Piyuancun Formation in Anhui Province of South China (as *Horodyskia* cf. *minor*[7]). However, it is debated whether *H. minor* should be classified as a species of *Horodyskia*[13,19,24]. For example, Calver et al.[13] excluded *Horodyskia minor* from the genus *Horodyskia* because the type species *Horodyskia mon-iliformis* has larger bead size (2.1–9.2 mm in diameter[23]) and lacks quartz in its halos.

Two recent studies have also removed the Liuchapo silicified specimens from the genus *Horodyskia* and assigned them to different genera. Yi et al.[11] reported similar "string of beads" specimens from the Liuchapo Formation in Hunan Province but classified them as *Longbizuiella hunanensis*, based on their smaller size and different preservational style from the Mesoproterozoic *Horodyskia*. Yi et al.[11] also re-assigned *Horodyskia minor*, *Horodyskia* cf. *minor*, and *Horodyskia* sp. fossils from South China[7–9] to *Longbizuiella hunanensis*. However, Yi et al.[11] synonymized *H. minor*, including its holotype, with *L. hunanen-sis*, effectively making *L. hunanensis* a junior synonym of *H. minor*. More recently, Liu et al.[12] erected another new genus—*Parahorodyskia*, with the type species *P. disjuncta* characterized by the absence of connecting filaments, and the new combination *P. minor* diagnosed with the presence of connecting filaments. Liu et al.[12] placed Liuchapo specimens without connecting filaments, including some specimens of *Longbizuiella hunanensis* in Yi et al.[11], in *P. disjuncta*, but enigmatically, these authors placed the holotype of *Horodyskia minor* reported in Dong et al.[8], which also lacks connecting filaments, in *P. minor*.

We note that neither the original diagnosis[6] nor the subsequent description of *Horodyskia*[23] excludes sub-millimetric beads for *Horodyskia*, and different types of preservation have been found in the same stratigraphic unit (e.g., the Shiwangzhuang Formation). The Shiwangzhuang material also confirms the co-existence of "string of beads" with two different bead size classes on the same bedding surface of argillaceous limestone (e.g., Figs. 2a, b, f; 6), as well as the observation that these fossils share a similar correlation between bead diameter and spacing (Fig. 6b), indicating that the two different size classes may represent congeneric species. Thus, *H. minor* fits the diagnosis of *Horodyskia*, and we agree with Dong et al.[8] that *H. minor* is a species of *Horodyskia*. Therefore, *Longbizuiella hunanensis* and *Parahorodyskia disjuncta* should be regarded as junior synonyms of *Horodyskia minor*.

Based on compilation of published specimens of *Horodyskia* (Fig. 6), we proposed that *H. minor* and *H. moniliformis* be differentiated on the basis of average bead diameter, with a cutoff size of 0.8 mm. The re-measured average bead diameter of *Horodyskia moniliformis*? specimens in Figs. 4.9 to 4.12 in Shen et al.[10] is 0.38–0.68 mm and the re-measured average bead diameter of *Horodyskia* cf. *minor* specimens in Fig. 3 in Dong et al.[7] is 0.10–0.67 mm. Thus, they are placed in *H. minor*. Luo and Miao[9] reported specimens of *Horodyskia* sp. from the Liuchapo Formation that have elliptical beads, which are ~0.05–1.5 mm in length. The specimens of *Horodyskia* sp. with small beads (e.g., specimens in Figs. 5A, B; 6A in ref.[9]) are considered as *H. minor*. However, the specimens with larger beads in Luo and Miao[9] include beads that contain smaller *Nenoxites*-like organisms (e.g., beads in Fig. 7F, I in ref.[9]). Whether these latter specimens can be classified as *H. minor*, *H. moniliformis*, or something else needs further examination.

*Parahorodyskia minor*, which was diagnosed with the presence of connecting filaments but was typified by a specimen lacking connecting filaments (i.e., the holotype of *Horodyskia minor*, Fig. 3a in Dong et al.[8]). Considering that the connecting filaments between beads are likely genuine biological structures, we suggest that Liuchapo "string of beads" specimens with connecting filaments, including specimens in Fig. 3c in Dong et al.[8] and specimens in Fig. 5J–L in Liu et al.[12], may represent a new species of *Horodyskia*.

## Discussion

The discovery of Mesoproterozoic "string of beads" has incited intense and long-standing debate regarding the biogenicity of these structures, with opposing viewpoints advocating that the "string of beads" represents "enigmatic structure", "dubiofossil", or "pseudofossil"[5,19–21]. *Horodyskia* fossils from the Shi-wangzhuang and Jiuliqiao formations of North China provide evidence to illuminate their biogenicity. Their carbonaceous nature (Figs. 4h–k; 5g–l), similar to representative carbonaceous compression macrofossils from the Shiwangzhuang Formation (Supplementary Fig. 2), suggests an underlying biological origin. Importantly, the carbonaceous compression specimens and three-dimensionally preserved organic-walled specimens are similar in morphology and size, strongly disputing the possibilities of sedimentary structures, mud flocs, and intraclasts. Raman spectroscopy reveals that the carbonaceous material of the Shiwangzhuang *Horodyskia* specimens have spectral char-acteristics similar to co-existing multicellular fossils, indicating that they both experienced low-grade metamorphism with apparent peak metamorphic temperatures [T(RmcRO%); *sensu* ref.[30]] (122–147 °C for six specimens of *Horodyskia* and 132–141 °C for three specimens of multicellular fossils; Supplementary Data 2). Given the presence of possible compressional folds and three-dimensional preservation, the Shiwangzhuang and Jiuliqiao specimens suggest that *Horodyskia* beads are bona fide fossils with organic-walled vesicles and they have suffered various degrees of postmortem collapse and diagenetic compaction.

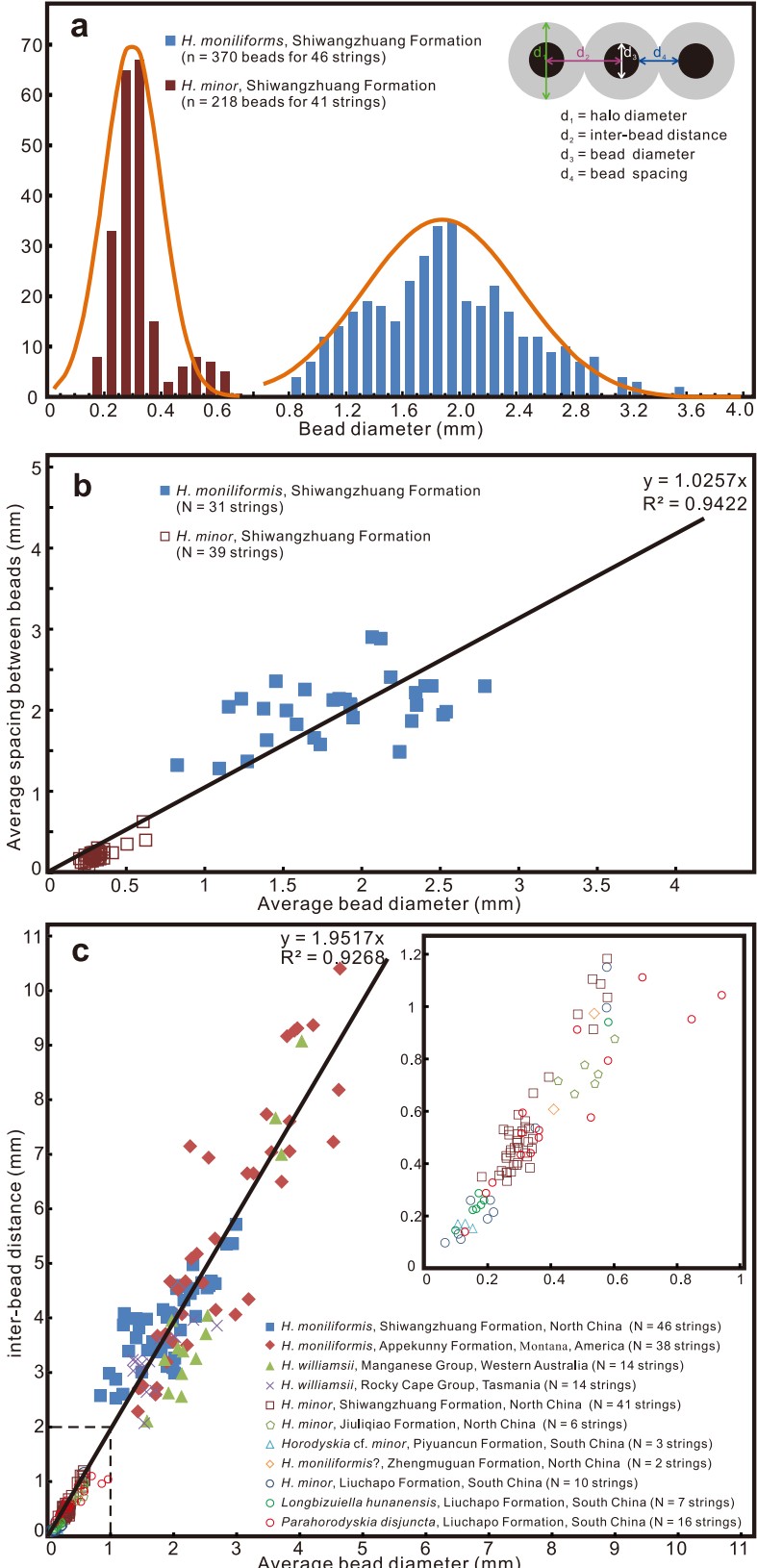

The consistent morphology, bead size distribution, and bead spacing of *Horodyskia* are inconsistent with the interpretation of *Horodyskia* as microbial mat fragments[19]. Rule and Pratt[19] recently proposed that *Horodyskia* is a pseudofossil formed by flocs and flakes trapped on protrusions of microbial mats, based on their observation of material from the early Mesoproterozoic Appekunny Formation. However, microbial mat fragments tend to be irregular in shape and their sizes would likely follow a power-law distribution[31]. In contrast, *Horodyskia* beads from the Shiwangzhuang Formation show a bi-modal size distribution

**Fig. 6 Biometric analysis of *Horodyskia*. a** Frequency distribution of bead diameter for Shiwangzhuang material; cartoon in upper right illustrates morphometric measurements in this study. **b** Cross-plot of average bead diameter versus average spacing between beads (i.e., gap between adjacent beads; *sensu* ref. [8]) of *Horodyskia* strings for Shiwangzhuang material. **c** Cross-plot of average bead diameter versus average inter-bead distance of *Horodyskia*-like strings from Shiwangzhuang and Jiuliqiao formations and previously published data from other localities (*Horodyskia moniliformis*, Appekunny Formation[5, 19, 23]; *Horodyskia williamsii*, Manganese Group[15, 25]; *Horodyskia williamsii*, Rocky Cape Group[13]; *Horodyskia* cf. *minor*, Piyuancun Formation[7]; *Horodyskia moniliformis*?, Zhengmuguan Formation[10]; *Horodyskia minor*, Liuchapo Formation[8]; *Longbizuiella hunanensis*, Liuchapo Formation[11]; *Parahorodyskia disjuncta*, Liuchapo Formation[12]); inset diagram is an enlargement of dashed box in lower left. See also Supplementary Data 1 for source data.

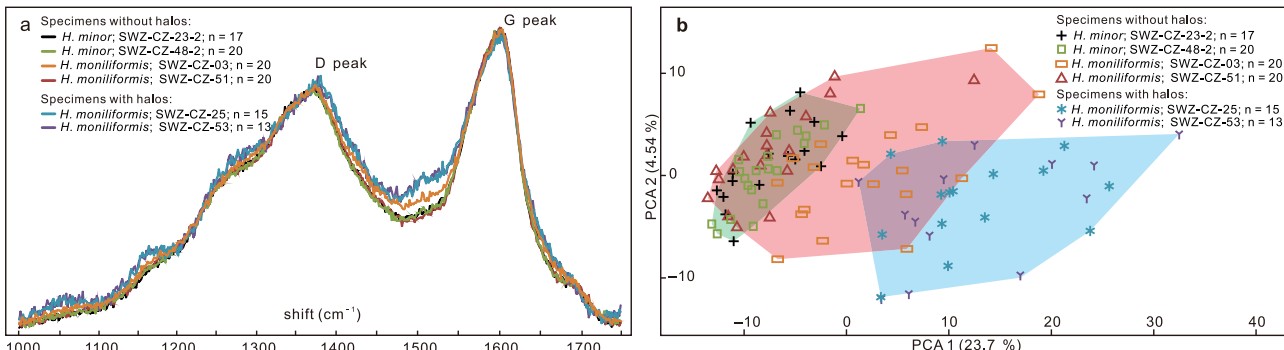

**Fig. 7 Raman spectroscopy of beads of four *Horodyskia moniliformis* specimens and two *Horodyskia minor* specimens from the Tonian Shiwangzhuang Formation. a** Average baseline-corrected and normalized Raman spectra of beads from six *Horodyskia* specimens. The beads of two *H. moniliformis* specimens (SWZ-CZ-25 and SWZ-CZ-53) are enveloped by halo, and the rest are not. Note that the six Raman spectra are not apparently different in D peak and G peak positions, nor in D/G peak intensity ratio. **b** Principal component analysis (PCA) of all baseline-corrected and normalized Raman spectra measured on six specimens, showing largely overlapping chemospace distributions among *Horodyskia* specimens without halos (red convex hull: *H. moniliformis*; green convex hull: *H. minor*), but marginally overlapping chemospace distributions between *Horodyskia* specimens with (blue convex hull) and without halos (red and green convex hulls). Letter "n" represents number of spectra collected from each specimen. See also Supplementary Data 2 for source data.

with a narrow average/standard deviation ratio (A/SD; 3.5 for *H. moniliformis* and 3.2 for *H. minor*; Fig. 6a), consistent with a biogenic assemblage of two species, rather than "biomorphs" which usually have unimodal size distributions with A/SD of 1.7–2.7 (ref. [32]). *Horodyskia* beads from other units also exhibit a bi-modal size distribution and a narrow A/SD ratio (Fig. 6c). In addition, vertical thin sections of the Appekunny material did not reveal the supposed connection between *Horodyskia* beads and pinnacle-like tufts or other protrusions on microbial mats[19]. Therefore, the evidence for interpretation of the Appekunny *Horodyskia* as "flocs and flakes" is problematic.

While its biogenicity has been in dispute over decades, the taxonomy of *Horodyskia* is also in a state of flux, largely because of different preservational styles and bead sizes. For example, *H. moniliformis*, which has millimetric beads and was erected on the basis of the early Mesoproterozoic Appekunny material preserved as casts and molds in siliciclastic rocks, has never been previously found co-existing with *H. minor*, which has sub-millimetric beads and was erected on the basis of the late Ediacaran Liuchapo silicified material. The much smaller size, younger stratigraphic age, and silicification preservational style of *H. minor* cast some doubts on its assignment to the genus *Horodyskia*[11–13,19].

The Shiwangzhuang material described herein provides a valuable link between the early Mesoproterozoic and late Ediacaran horodyskids (*Horodyskia* and *Horodyskia*-like fossils). Two groups of fossils with distinct bead sizes are observed on the same bedding plane in the Shiwangzhuang Formation (e.g., Fig. 2a, f). They have the same preservational style, exhibit similar positive correlations between bead diameter and spacing (Fig. 6b) and between bead diameter and distance (Fig. 6c), and share a largely overlapping chemospace distribution (Fig. 7b), suggesting that they are not only biogenic, but also likely congeneric organisms. The ranges of their bead diameters

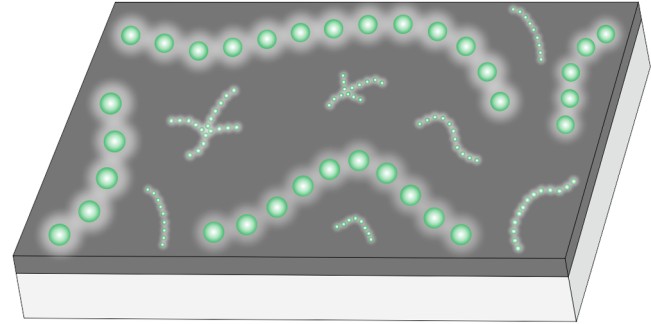

**Fig. 8 Schematic reconstruction of *Horodyskia moniliformis* (larger beads) and *Horodyskia minor* (smaller beads).** Reconstructions are based on specimens from the Tonian Shiwangzhuang and Jiuliqiao formations in North China. The co-occurrence of *H. moniliformis* and *H. minor* on the same bedding plane is based on specimens illustrated in Fig. 2a, f from the Shiwangzhuang Formation.

overlap with that of *Horodyskia minor* from the Liuchapo Formation and that of *H. moniliformis* from the Appekunny Formation, respectively (Fig. 6c), supporting the assignment of specimens with sub-millimetric beads to the genus *Horodyskia*. Therefore, our study suggests that *Horodyskia* has a stratigraphic range from early Mesoproterozoic to late Ediacaran, with remarkable long-term morphological stability over 900 Myr.

Marginal thickenings (e.g., Figs. 2l; 3f–h) and concentric rings (e.g., Fig. 3i), interpreted as possible compressional folds, are commonly observed on the beads of Shiwangzhuang *H. moniliformis* specimens, providing evidence for their originally spheroidal morphology prior to compaction. As a result of taphonomic compression, such folds are common in

carbonaceous compressions of spheroidal organic-walled vesicles[33]. These compressional folds, together with the typically round to oval morphology of the beads that can be sometimes preserved three-dimensionally, indicate that *Horodyskia* beads were originally spheroidal in shape and had a recalcitrant organic wall. The inferred spheroidal shape is consistent with the studies of *Horodyskia* materials from Western Australia[15,25] and South China[8,12], but is in contrast to the "wide ice cream cones" reconstruction of *Horodyskia* based on the Appekunny material[23]; the latter reconstruction has also been questioned in subsequent restudy of the Appekunny material[19,24].

Overall, *Horodyskia* can be reconstructed as a chain-like organism or a colony of individuals composed of several to dozens of vesicles or cells with a recalcitrant organic wall and likely embedded in amorphous gelatinous matrix (Fig. 8). The strings of *Horodyskia* are randomly orientated and are not superimposed on each other, suggesting that *Horodyskia* was probably a procumbent and epibenthic organism laying on sediment surface, and the specimens do not seem to have been allochthonously transported. They do not have a specialized holdfast structure, nor is there evidence for morphological differentiation, consistent with a procumbent and epibenthic life style, rather than an erect benthic or a pelagic life style.

Various phylogenetic interpretations have been proposed for *Horodyskia*, including impression of *Scaberia*-like brown alga with serially distributed float bladders[15], nodose and branching *Armstrongia*-like sponge[22], tissue-grade colonial eukaryote and possible metazoan[23], archaeal-bacterial consortium[2], agglutinated foraminifer[8,12], bladder-like cell of *Geosiphon*-like fungus[24], testate amoeba[9], and green alga[12]. Grey et al.[25] had a critical review on all the phylogenetic interpretations of *Horodyskia* proposed prior to 2010 by paleontologists. However, the authors in Grey et al.[25] failed to reach a consensus, although they favored the seaweed and hydrozoan interpretations. *Horodyskia* fossils presented in this study provide an opportunity to reappraise these various phylogenetic interpretations.

Morphological evidence indicates that *Horodyskia* is unlikely a prokaryotic organism. The observed division in beads and paired beads likely resulted from binary fission indicate that the submillimetric to millimetric beads of *Horodyskia* very likely represent giant cells. Although the inferred binary fission of the *Horodyskia* cells is similar to transverse cell division in filamentous bacteria such as oscillatorialean cyanobacteria[34], *Horodyskia* cells are orders of magnitude larger than typical prokaryotic cells, and their inferred recalcitrant cell walls with a large size are unusual for prokaryotic cells[2,35]. We note that some modern sulfur-oxidizing bacteria, e.g., *Thiomargarita*[36,37], can reach millimeters in cell size, but their cell construction is fundamentally limited by diffusion; as a result, their cells contain large metabolically inactive vacuoles and/or adopt an elongate morphology to maintain a physiologically viable ratio of surface area to volume. The volume of the spherical cells of *Thiomargarita namibiensis* ($\leq 2.2 \times 10^8$ $\mu m^3$; Table S2 in ref. [38]) and long tubular cells of Ca. *Thiomargarita magnifica* ($\leq 2.2 \times 10^7$ $\mu m^3$; Table S2 in ref. [37]) is two to three orders of magnitude less than the volume ($\leq 2.3 \times 10^{10}$ $\mu m^3$) of the beads of *Horodyskia*. Importantly, no modern sulfur-oxidizing bacteria are known to have recalcitrant cell walls. Therefore, considering its extremely large cell size and inferred recalcitrant cell wall, *Horodyskia* is unlikely to be a prokaryotic organism, although we cannot entirely rule out the possibility that some extinct lineages of prokaryotes could have developed such a large cell size and cell-wall recalcitrance[39].

The large cell size of *Horodyskia* indicates that it is not only a eukaryote, but also likely a multinucleated or coenocytic eukaryote. The sub-millimeter- to millimeter-sized cells of *Horodyskia* seem to require multiple nuclei to regulate the giant mass of cytoplasm, because a single nucleus, via the diffusion of messenger RNAs, can only control a limited volume of cytoplasm[1,40,41]. The cellular nature and the unconnected but occasionally dividing feature of *Horodyskia* beads, indicate that these fossils are unlikely to be complex multicellular organisms such as articulated brown alga[15], branching sponge[22], tissue-grade colonial metazoan[23], or endocyanotic fungus[24], but are more likely to be protists whose clonal cells forming simple and not fully integrated colonies (simple clonal coloniality; *sensu* ref. [1]). Multicellularity (including clonal and aggregative development) occurs in a number of eukaryotic groups, including fungi, animals, choanoflagellates, slime molds (dictyostelids, myxomycetes, protostelids, and acrasids), green algae, land plants, red algae, ciliates, oomycetes, diatoms, chrysophytes, xanthophytes, and brown algae[42–44]. However, protists with a construction of simple clonal coloniality, a sub-millimetric to millimetric cell size (or inferred coenocyte), and a recalcitrant cell wall, features that define *Horodyskia*, are limited to a smaller number of eukaryotic clades[45]. Here we will evaluate the following three most likely potential analogs: arcellinid testate amoebae[46], foraminifers[45], and some algal groups[47–49], which are proposed in previous studies and show some, if not all, of the features of *Horodyskia*.

Recently, arcellinid testate amoebae have been suggested as potential analogs of *Horodyskia*[9], although they are exclusively unicellular, not colonial[42,44]. Indeed, some arcellinid testate amoebae possess an organic-walled test[45] that has the potential to be preserved as microfossil[50]. However, the lack of an aperture in the beads of *Horodyskia* argues against an affinity with arcellinid testate amoebae, which typically possess a characteristic aperture. The sizes of arcellinid testate amoebae, commonly between 50 μm and 200 μm in shell length[51], are an order of magnitude smaller than *H. moniliformis* beads and also smaller than most of the *H. minor* beads. So far, vase-shape microfossils (VSMs) from the late Tonian Period are widely accepted as the earliest fossilized arcellinid amoebae[51]. Phylogeny of extant arcellinid amoebae and VSMs based on ancestral-state reconstructions put the divergence of major arcellinid lineages at ~759–734 Ma and molecular clock estimate using VSMs as a calibration places the divergence time at ~1000–730 Ma[46,52,53], much younger than the oldest known *Horodyskia* fossils from the ~1.48 Ga Appekunny Formation[23].

Some rhizarians, particularly foraminifers, have also been proposed as modern analogs of *Horodyskia*[8], although rhizarians are also exclusively unicellular[42,44]. The halos, beads, and connecting filaments of *Horodyskia minor* (or "*Parahorodyskia minor*") were interpreted as, respectively, agglutinated tests, cytoplasm-filled chambers, and cytoplastic passages of foraminifer cells[8,12]. Importantly, many modern foraminifers have an inner organic enveloping structure (i.e., organic lining) that can be preserved as organic-walled microfossil[54], and some foraminifers (e.g., monothalamids) even have an entirely organic test[55], although with a limited fossil record[56]. The Xenophyophorea, a group of deep-sea foraminifers also belonging to the monothalamids, can reach a millimetric to centimetric size and are multinucleated and they are among the largest unicellular rhizarians[57]. Some xenophyophores (e.g., *Aschemonella monile*) also possess a multi-chambered test resembling closely packed "string of beads"[58]. Thus, it is possible that the carbonaceous wall of *Horodyskia* beads may represent organic linings or organic tests of single-chambered monothalamid foraminifers, and hence the string of beads would represent either a multi-chambered foraminifer[8,12] or, less likely, a colonial rhizarian. Still, the lack of an aperture in *Horodyskia* beads presents an obstacle to the foraminifer interpretation. In addition, the earliest unambiguous foraminifer fossils are of early Cambrian in age[59,60], and a molecular clock study suggests that foraminifers diverged at

~920–650 Ma[61], much younger than the oldest known *Horodyskia* fossils from the ~1.48 Ga Appekunny Formation[23].

An alternative and perhaps better interpretation is that *Horodyskia* may represent a giant-celled and colonial alga. Giant-sized cells, which are often coenocytic, occur in five extant algal clades, including Xanthophyceae, Rhodophyceae, Chlorophyceae, Charophyceae, and Ulvophyceae[40,47,48]. In the Xanthophyceae, Vaucheriaceae (e.g., *Vaucheria* and *Botrydium*) is the only family that develops giant cells[40,49]. *Vaucheria* has a branching and tubular thallus with a holdfast, and *Botrydium* has large vesicles and branching rhizoidal extensions[49]. In the Rhodophyceae, *Griffithsia* is the only genus that develops giant cells, and it possesses a heavily branched thallus with a holdfast[40,49]. The Chlorophyceae, Charophyceae, and Ulvophyceae also contain giant-celled members[47,62], and paleontological and molecular clock data do suggest a Mesoproterozoic origin for the "core Chlorophyta"[63,64]. Worth mentioning is the extant ulvophycean *Valonia ventricosa* (=*Ventricaria ventricosa*), which is characterized by spherical giant cells that are millimetric to centimetric in size and bear reduced rhizoids[65]. Individually, these giant algal cells resemble *Horodyskia* beads in size and shape, but they do not form uniseriate chains embedded in a gelatinous matrix. This difference, along with the lack of rhizoids in *Horodyskia* beads, makes the giant-celled algae yet an imperfect modern analog for *Horodyskia*. Nonetheless, it is also reasonable that we would not expect an evolutionary stasis experienced by eukaryotes for ca. 1.5 billion years and a parsimonious interpretation is that *Horodyskia* represents a total-group multicellular eukaryote that was composed of a string of giant-sized cells (probably coenocytic) and achieved a macroscopic size in the Proterozoic Eon.

Achieving a macroscopic body size occurred multiple times through multicellularity or coenocytism in different eukaryotic clades[1,40,66], and this has important evolutionary, ecological and geobiological implications. The Shiwangzhuang and Jiuliqiao materials indicate that *Horodyskia* acquired a large body size visible to the naked eyes probably through the combination of coenocytism and simple clonal coloniality. A coenocytic body plan has been proposed as the direct progenitor of some multicellular algal, animal, and fungal groups, through subsequent process similar to segregative cell division, whereas a colonial body plan is traditionally regarded as progenitor of multicellular organisms[67–69]. It is interesting to note that *Horodyskia* may have stood at an evolutionary crossroad of the "coenocytic-to-multicellular" and "unicellular-to-colonial-to-multicellular" transformation series[69], and seems to represent a primitive condition for siphonocladous (multicellular multinucleate)[68]. Macroscopic body sizes can bring noticeable ecological advantages to eukaryotes, including increased speed and efficiency to occupy new adaptive niches or migration to more favorable environment, better protection against phagocytic predation, and increased possibility to capture larger preys[70–72]; larger eukaryotes can also act as a faster biological pump by accelerating the sinking flux and then enhancing the efficiency of organic carbon burial, and therefore facilitate marine ventilation[71,73,74]. *Horodyskia* is among the few examples of macrofossils reported from the early Mesoproterozoic[23]. Considering that the earliest *Horodyskia* fossils are from the ca. 1.48 Ga Appekunny Formation in Montana[23], our study implies that macroscopic giant-celled (probably coenocytic) protists existed in the early Mesoproterozoic. Other examples of early Mesoproterozoic or earlier macrofossils include the helical fossils *Grypania* and *Katnia*[75,76], the tomaculate fossil *Tawuia*[75], and unnamed carbonaceous fossils from the 1.56 Ga Gaoyuzhuang Formation of North China[77], all of which have been known as carbonaceous compressions. Together, these fossils provide paleontological evidence for the evolution of macroscopic size in the early-middle Proterozoic and it is worth evaluating the possibility that they may have also been coenocytic organisms[73].

The late Paleoproterozoic to early Mesoproterozoic era witnessed the early diversification of eukaryotes, as represented by organic-walled microfossils with complex ornamentations, e.g., *Valeria*, *Tappania*, *Shuiyousphaeridium*, and *Dictyosphaera*[78,79]. Late Paleoproterozoic to early Mesoproterozoic macrofossils mentioned above, including *Grypania*, *Katnia*, *Tawuia*, and *Chuaria*, may also be eukaryotes[73,76,80] (however, see refs. [75,81,82]). The interpretation of *Horodyskia* as a colonial giant-celled protist (probably coenocytic), considering its oldest occurrence dating back to ~1.48 Ga[23], adds to a short but growing list of early eukaryote fossils existed in this critical time interval, and suggests that a macroscopic coenocytic body plan has probably been present since the early Mesoproterozoic.

It is important to point out that the Shiwangzhuang and Jiuliqiao formations also preserve other giant-celled fossils that have been interpreted as possible macroscopic coenocytic algae, including the annulated tubular fossils *Sinosabellidites*, *Protoarenicola*, and *Pararenicola*, the multicellular trichomes *Anqiutrichoides* and *Eosolena*, and the tomaculate fossil *Tawuia*[73,83,84]. Among them, *Eosolena magna* is somewhat similar to *Horodyskia* in its sub-millimetric to millimetric cells that are discoidal to isodiametric in shape. Together with the "string of beads" fossil *Horodyskia* herein and the siphonocladous *Proterocladus* from the early Tonian Nanfen Formation in North China[63], these North China materials suggest that giant-celled protists may have been diverse in the Tonian Period when they became increasingly important in the paleoecology and geobiology during this time interval[73,83].

In conclusion, on the basis of the Shiwangzhuang and Jiuliqiao materials, *Horodyskia* is reconstructed as a benthic chain-like organism consisting of several to dozens of cells with a recalcitrant organic wall, and the cells are likely embedded in amorphous gelatinous matrix. Our study reinforces the biogenicity of *Horodyskia* through the detailed characterization of carbonaceous compression specimens from the Tonian successions in North China. The Tonian fossils herein also provide a valuable link, both chronologically and morphologically, between early Mesoproterozoic and late Ediacaran horodyskids insofar as two species with distinct bead sizes co-occur on the same bedding plane in the Shiwangzhuang Formation. The most plausible phylogenetic interpretation for *Horodyskia* is that it represent a multicellular and giant-celled protist, which has acquired a large body size and shares some similarities with living coenocytic algae and monothalamid foraminifers, although these two groups are phylogenetically distant and the latter are typically unicellular and therefore less likely. Our study reports evidence that *Horodyskia*, a genus of macroscopic fossils ranging from the early Mesoproterozoic Era to the terminal Ediacaran Period, may have attained its macroscopic size through the combination of coenocytism and simple clonal coloniality. Together with other possible early-middle Proterozoic coenocytic fossils[73], *Horodyskia* provides an important temporal constraint on the origin of coenocytic eukaryotes.

## Methods

**Studied material**. Specimens were collected from an argillaceous limestone horizon ca. 74 m below the top of the Shiwangzhuang Formation (~850–720 Ma) at the Baishicun section (36°30′39″N, 119°07′39″E), Anqiu region, western Shandong Province, North China (Fig. 1b) and an argillaceous limestone horizon at ca. 14 m below the top of Jiuliqiao Formation (~950–720 Ma) at the Baiguashan section (32°44′24″N, 117°11′24″E), Huainan region, northern Anhui Province, North China (Fig. 1c). All illustrated specimens from the Shiwangzhuang Formation are deposited in College of Earth Science and Engineering, Shandong University of Science and Technology and illustrated specimens from the Jiuliqiao Formation are

deposited at Nanjing Institute of Geology and Palaeontology, Chinese Academy of Sciences (NIGPAS). Further information and requests for materials should be directed to K.P. (kepang@nigpas.ac.cn).

**Light microscopy (LM)**. Specimens were examined and photographed using a Nikon D810 digital camera, an Olympus szx16 stereomicroscope attached with a DP74 digital camera, and a Zeiss Axioscope A1 microscope attached with an Axiocam 506 digital camera. Diameters of the bead and halo were averaged between the maximum and the minimum dimensions. Both the distance between the centroids of adjacent beads (inter-bead distance)[25] and the spacing (or gap) between adjacent beads[8] were measured to overcome the uncertainty of beads with fuzzy and irregular edges[25].

**Scanning electron microscopy (SEM)**. Well-preserved specimens were selected for observation using both backscattered electron (BSE) and secondary electron (SE) detectors and elemental mapping using energy dispersive X-ray spectroscopy (EDS) on a TESCAN MAIA 3 GMU field emission scanning electron microscope (SEM) in Nanjing Institute of Geology and Palaeontology, Chinese Academy of Sciences (NIGPAS). The operating voltage in BSE-SEM was 20 kV in high vacuum conditions.

**Raman spectroscopy**. Selected specimens were analyzed on a Horiba HR Evolution Jobin Yvon Raman microprobe in Nanjing Institute of Geology and Palaeontology, Chinese Academy of Sciences (NIGPAS). The Raman microprobe is equipped with a 532 nm argon laser source and a 600 grooves/mm grating. The laser beam was <2 μm in diameter and a 50× objective lens was used to collect backscattered radiation. Spectra in the range of 800–2000 $cm^{-1}$ were acquired using the software LabSpec 6.0 with exposure time of 40 or 60 s and averaged after three times of acquisition[85]. At least thirteen spots were tested for each specimen. The spectral range of 1000–1750 $cm^{-1}$ was used for baseline correction using the PeakFit 4.2 software and then normalized using the formula: normalized intensity = (X–Min)/(Max–Min), where the Min and Max represent, respectively, the minimum and maximum intensities of each spectrum[85]. The baseline-corrected and normalized Raman data were subsequently subjected to principle component analysis (PCA) in JMP13. The chemospace of carbonaceous material were defined by the first and second principal components from the PCA analysis[85].

**Reporting summary**. Further information on research design is available in the Nature Portfolio Reporting Summary linked to this article.

## Data availability
All data needed to evaluate the conclusions in the paper are present in the paper and its supplementary information files, or from K.P. (kepang@nigpas.ac.cn) upon reasonable request. Specimens from the Shiwangzhuang Formation illustrated in this paper are reposited and available at College of Earth Science and Engineering, Shandong University of Science and Technology and specimens from the Jiuliqiao Formation illustrated in this paper are reposited and available at Nanjing Institute of Geology and Palaeontology, Chinese Academy of Sciences (NIGPAS). Source data are provided with this paper. The catalog numbers for the specimens are provided in Supplementary Data 3.

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

## Acknowledgements

This research was supported by the National Key R&D Program of China (2022YFF0802700), National Natural Science Foundation of China (42202008, 41921002,

42272005, 42192501, and 42272001), Chinese Academy of Sciences (XDB26000000), State Key Laboratory of Palaeobiology and Stratigraphy (20201102 and 213108), Youth Innovation Promotion Association of CAS (2021307), China Postdoctoral Science Foundation (2021M693243) and Taishan Scholars Project (tsqn201812069). S.X. was supported by the U.S. National Science Foundation (EAR-2021207). We would like to thank Nick Butterfield, Leigh Anne Riedman, and Veeru Kant Singh for constructive comments. Xiaofeng Xian, Yunpeng Sun, Lei Zhang, Shengong Zhang, Jing Fang, and Zhengqi Zhao are acknowledged for assistance in field work.

## Author contributions

K.P. is the lead contact for this paper. K.P. and L.C. designed the study. G.L., C.W., and K.P. conducted fieldwork. G.L. and K.P. conducted microscopic observations and Raman spectroscopy. K.P., S.X., G.L., L.C., X.Y., and Q.T. developed the interpretation. G.L. and K.P. prepared the initial draft of the manuscript with input from S.X., Q.T., C.Z., and all the other authors.

## Competing interests

The authors declare no competing interests.
