## [Peer Review File · Communications Biology]

Reviewers' comments:

Reviewer #1 (Remarks to the Author):

Comments on: Tonian carbonaceous compressions shed light on Horodyskia, one of the oldest multicellular and coenocytic macro-organisms

The authors present new materials of *Horodyskia moniliformis* and *H. minor* from the Tonian Shiwangzhuang and Jiuliqiao formations and present data based on carbonaceous compressions to suggest a coenocytic eukaryotic body plan for these taxa. *Horodyskia moniliformis* is a well known but enigmatic macrofossil ranging from the early Mesoproterozoic to the end of the Ediacaran; insights to its biological affinity and mode of life are of great interest not only to Precambrian paleontologists, but also evolutionary biologists as both groups grapple with the origin and early evolution of eukaryotes and the pathways to macroscopic size. This paper presents interesting new fossil data and an analysis of body plan and affinity hypotheses. I recommend publication of this manuscript after rather minor revisions and development.

General:

- There are several opportunities to shorten and tighten this manuscript that will benefit the reader and the material. I mention some of these in my line-by-line comments below.
- There's mention of additional multicellular fossils and reference to comparing Raman data of those and the *Horodyskia* specimens- those data need to be shared, at least in the supplementary files.
- I find myself wanting more morphological analyses of the carbonaceous remains- Do you get anything useful from SEM of the beads? Can you see wrinkles or ripples to the vesicle?
- Modern testate amoebae are found in several very distant clades, most prominently the Amoebozoa and the Rhizaria. Rhizarian testate amoebae do not have a well documented deep fossil record; almost all fossil testate amoebae belong to the Amoebozoa. I have pointed out changes that need to be made to accommodate this.
- Although I think it's technically ok since there are not genera or species erected here, having nomenclatural acts such as emendations in the supplementary files should be avoided.

Line-by-line:

Lines 21-22- You can delete these first two sentences for a stronger start. It isn't necessary to explain why macroscopic fossils are important and the first sentence doesn't actually convey any information. The second sentence has a few problems- what do you mean by "unambiguous macrofossils"? It isn't that their macro- status ambiguous. I'd guess you mean that their affinity is ambiguous- but that isn't clear here. Secondly, the phrase "earliest crown-group eukaryotes" needs conditions placed on it- those around 1 Ga are the earliest fossils that have been allied in the literature with crown groups, but two things: 1) if those fossils are representatives of crown groups- we can't say that earlier crown group eukaryotes didn't exist- certainly they must have, in fact; and 2) those are interpretations that those ~1Ga fossils are crown-group and even if that interpretation is generally accepted, it is still an interpretation and that needs to be stated. However, the larger point here is that you could cut those first two sentences and get right to the point with talking about *Horodyskia*. E.g. start with "*Horodyskia* is one of the few..." and you might consider combining it with the next sentence with a "but the biological interpretation..."

Line 27- purely stylistic, but if you chose to say "has been", you should give a since of time, like "the biological interpretation of this enigmatic fossil, however, has been a matter of controversy since its initial description in 2000". Otherwise you might change 'has been' to 'is'. Incidentally, in looking to remind myself of when *Horodyskia* was first described, I noticed that you don't include Yochelson and Fedonkin (2000) in your references. If you have space for its inclusion I very much recommend you

include it.

Lines 27-30- Since the beginning of this abstract set up the fact that there is controversy about the biological interpretation of Horodyskia, the reader expects the next few sentences to indicate that you will address that matter with your materials.

Line 30- can you be more explicit about what you mean by "precious link"?

Lines 31-34- I'm not sure what you're wanting to do with this sentence- basically it is saying that Horodyskia is interpreted as a coenocytic eukaryote and it is old and long ranging. This thought needs to be more developed here- what about the existence of coenocytic eukaryotes at 1.5 Ga will help with our understanding of eukaryotic evolution or the evolution of a multicellular form? Additionally, the sentence structure is somewhat difficult to parse- is there a reason we would exclude or not take into consideration the specimens from Montana (the type specimens)?

Line 40- It's generally not acceptable in formal writing to use "but" as the first word in a sentence. You could side-step this by instead using "However".

Line 43: you might add the word "evolution" at the end of this sentence. It's not really the tempos and modes of individual early eukaryotes that we're interested in here, but the tempo and mode of early eukaryotic evolution.

Line 45- you might consider switching the ages here to move from older to younger. It makes more sense that the stratigraphic range extends from the early Mesoproterozoic to the terminal Ediacaran.

Line 45- Rather than "It is", you might say "This genus is" to be more precise in wording.

Line 51- I appreciate having more insight on the likely age of the Backdoor and Stag Arrow fms, but you need to include a reference for why the age is probably 1.21-1.07 Ga rather than the other age range you state.

Line 56- there's no reason for the wide global distribution and long stratigraphic range to inform on the biological affinity of Horodyskia, so the "in spite of" is a little odd here. You might get the same idea across, i.e., that even though we have lots of specimens to study, we still don't know who Horodyskia is, by simply saying, "Horodyskia has a wide global distribution and long stratigraphic range but its biological affinity remains controversial..." or if you really want to keep "in spite of", you should make the point that its in spite of the many specimens to study, not in spite of the stratigraphic range and global distribution.

Line 57-58- I like the phrase, "even the debate on its biogenicity swings back and forth."

Line 59- it seems that pseudofossil and dubiofossil should be mentioned back-to-back here.

Lines 62-70- This! This needs to be mentioned in the abstract- boil down to one sentence and that can be somewhere near the end, followed by quick results. It is a big deal that you're reporting organic-walled specimens of Horodyskia and that is something that sets this work apart from previous attempts to discern Horodyskia's affinity.

Lines 73-77- what do you mean by "constraints"- age constraints? This wording is vague and needs some attention. What precisely do these fossils offer in terms of a better understanding of origins of multicellularity, the early evolution of multicellularity or the origin or early evolution of the coenocytic

body plan. And what does knowing there were multicellular organisms or coenocytes in the Mesoproterozoic tell us? Try to be specific about what your fossils can tell us (and tell us what they can't say!) rather than using vague and imprecise terms.

This sentence makes a bit more sense here than it does in the abstract because it has more context, but the problem remains as to why one would need to exclude the material from Montana- 1) why those fossils? Because they're very old and there's a concern about the affinity interpretation being extended to 1.48 Ga? 1a) then that means you're questioning the identification of your materials as Horodyskia, as the fossils from Montana are the type materials and standard bearers. 1b) if it is about the age, why not be concerned about extending the affinity to fossils of Western Australia and Tasmania that are also Mesoproterozoic in age? 2) why would the age matter? If those are all really the same biological entity, i.e., all specimens you've mentioned in lines 48-55 are Horodyskia and it is monophyletic, then they should all have the same biological affinity. So this is a strange sentence- if there's a reason to consider the Montana fossils as separate, that needs to be discussed and the taxonomic situation needs to be sorted. If you're just trying to make the point that a coenocytic body plan dates to at least as old as 1.48 Ga, then say that.

Line 82- I would say it's not necessary to say "morpho- species". It's understood that the species are distinguished by morphological characters and there is the general assumption about biological relatedness applied to all fossil species, so unless there is good reason to think that Horodyskia is polyphyletic, then you could just say "species".

Line 85- I find the idea of having nomenclatural acts, including emendation of genera and species, in supplementary files very worrying. I think as long as the supplementary files are hosted as pdfs with their own ISSN or ISBN, then they may be acceptable as "effective publication" according to the International Code of Nomenclature for Algae, Fungi and Plants- and I guess since you're not naming new genera or species here it won't cause a problem, but I'd expect that the emendation will not get the notice and application it otherwise would get if published in main text. For reference, here is a link to one of the key sections on effective publication.

(https://www.iapt-taxon.org/nomen/pages/main/art_29.html)

Lines 114-115- The phrase "which also occur in aggregation" is out of place here- it seems like you're saying that the Jiuliqiao Formation also occurs in aggregation.

Line 147- The chemospace discussion needs more of a set up before you discuss results- just one additional sentence should suffice.

Line 161- please say more about the coexisting multicellular fossils- what taxa? Have they been published? If so, give the reference; if not maybe put them in supplementary files. -Ah, you do mention them. Maybe for the reader's benefit put a parenthetical here saying something like (briefly discussed below)

Line 164- If you're going to refer to data from those multicellular fossils, you need to include those data and information about the multicellular fossils.

Line 168-169- I'm not sure why you're repeating the shared thermal history point here. It's already been said and this sentence is otherwise about interpretations of the organic walls, which needs a bit more development.

Line 189-191- If this is the first co-occurrence of these two species, that should also be mentioned in the abstract when you mention that it is the first occurrence of organic-walled preservation.

Line 203- I think I might put in a "likely" here. I.e., "... suggesting that they are not only biogenic but also likely congeneric organisms". That is not a critical recommendation, but I, personally, would put it in.

Line 204- if these taxa were all synonymized, please cite reference.

Line 295- this seems to suggest that arcellinid testate amoebae are rhizarians, which is not accurate. Arcellinids are within the Amoebozoa and rhizarians are a branch within SAR; they are quite separate clades. Perhaps this is not what you meant and its just unfortunate sentence construction- this can also be read to say that there are coenocytic algal groups within the Rhizaria- I am not an expert on modern rhizarian diversity, but I am unaware of coenocytic photosynthesizing rhizarians so it might benefit your readers for you to both reword this sentence and to also expand it to give more details on these living groups.

Line 298- regarding the references- it is true that rhizarians can make organic tests and Adl et al do mention that, but the paper by Porter and Riedman is only about arcellinid testate amoebae, which within the Amoebozoa, a completely separate group from the Rhizaria. Additionally, a quick look at the cited work of Kutluk and Mazei shows fossil arcellinids (Amoebozoa), not rhizarian testate amoebae.

Lines 297-300- the idea of Horodyskia as testate amoebae has never made sense to me- the testate amoebae (Luo and Miao focus on those shaped like Arcella), but these testate amoebae are so small- like you mention in your paper, they are 50 to 200 microns wide (although at least the fossil testate amoebae tend to be much smaller than 200 μm , often less than 100 μm)- they are so small- an order of magnitude smaller than Horodyskia moniliformis beads and a half to less than a third of the diameter of H. minor. Aside from the size problem, testate amoebae, fossil or modern, don't form strings. Some Horodyskia have stolons connecting them- what would that be if these were testate amoebae? And yes, the lack of an aperture is an issue, but Luo and Miao focused on Arcella-type testate amoebae, so the aperture wouldn't be prominent. It is just such a strange argument, it seems like it derails your paper to present this hypothesis so prominently. Obviously you have to address this hypothesis, but what might help is to signal to the reader that in the following paragraphs you will be evaluating recent hypotheses of affinity- maybe with a subheading or just a sentence or clause before launching into it.

Line 302- This sentence needs to be changed to accommodate the fact that the testate amoebae discussed in the preceding sentence are not rhizarians.

Line 308-319- I don't have the answer for this, but in reading this passage I wonder what the occurrence of say, monothalamids, in the Mesoproterozoic would require in terms of other crown groups that would have had to appear before 1.48 Ga. How derived are these groups you mentioned? By 1.48 Ga we would have to have already evolved (that we have ?no? fossil evidence of?)

Line 318- It occurs to me that we probably need to use a skeptical eye when considering some molecular clock dates that use Bangiomorpha as a calibration point, but that predate its more tightly constrained age reported by Gibson et al (2017) that is ~200 million years younger than previously reported. And I would be particularly concerned about the older age reported by Pawlowski et al (2003) since in that paper they also used as a calibration point, some VSMS from the Chuar Group (Porter et al, 2003) that were even then only tentatively allied with rhizarian testate amoebae (Imbricatea). Since publication of that VSM paper in 2003, that once tentative suggestion of Rhizaria in the Tonian has not been supported and is thought to be less likely for various reasons. The point of this is that those ages for the emergence of the Foraminiferan Phylum are likely too old based on calibrations we now consider wrong.

Line 330-336- How do these algae reproduce? Is there any part of their lifecycle that could look like a dividing Horodyskia bead? Do you find multiple bubbles of Valonia or other algae in association with each other, if not actually connected and in a 'string'?

Line 335- More precisely, these specific modern, extant, coenocytic algae are imperfect matches for Horodyskia. It might be worth making the point that we don't have to find an exact match, do we really expect those species to persist unchanged for 1.5 billion years?

Line 347-351- This is an unnecessarily complex sentence. It seems like you're trying to say that the new data collected from the Shiwangzhuang specimens suggests a coenocytic nature for Horodyskia, a fossil genus that dates from the early Mesoproterozoic. I like that you're then bringing in to the discussion some of the other microfossils of the early Proterozoic; it is worth considering if they also might have this body plan- what should one look for in those taxa that would suggest this?

Line 360-361- These taxa are known from more units than just the Ruyang and Roper, so it's a little odd to mention only those two. You could just cut the unit names (but keep the citations) since your point is about time, not the specific units.

Line 364- This is the same point I've made before, but Why would you exclude the Appekunny Fm? I don't understand the point of including this sentence. If you're just trying to hedge your bets, don't. Just be full-throated about saying that your data suggest that Horodyskia was a macroscopic coenobium, so the macroscopic coenocytic body plan was present in the Mesoproterozoic (and probably since then- I say probably because that body plan likely has evolved repeatedly).

Line 383- be specific about what the "valuable link" is- is it that you're seeing the two taxa co-occurring?

Line 382- you mention detailed characterization of the carbonaceous compressions, and you do give chemical data, but can you give more morphological data? Can you see evidence of wrinkles or ripples in the vesicle if you look with SEM?

Line 387-392- This last sentence can be made much shorter and to-the-point. What you might do is to cut the beginning about the Appekunny and start with "Our study reports evidence that Horodyskia, a genus of macroscopic fossils dating from the early Mesoproterozoic to the terminal Ediacaran, attained its macroscopic size through the combination of coenocytism and simple clonal coloniality. This fossil and its body plan provide an important constraint for the origin and early evolution of coenocytic eukaryotes."

Regarding: Simple clonal coloniality. This wasn't discussed, so I can't see how you can argue that simple clonal coloniality contributed to Horodyskia's large size. Also, I'm not sure coloniality is a word.

Regarding: constraint for the origin and early evolution of coenocytic eukaryotes. This is still a vague statement and repeating it just makes the reader more aware of how vague it is. What is being constrained? Timing? What about early eukaryotic evolution can be better understood in light of these new data? Is there something inherent in this body plan that will allow an organism to do something it otherwise couldn't? to exploit a niche/live somewhere otherwise unavailable? Would a coenocytic organism have an effect on other organisms or the environment around it that some other style of macro-body plan wouldn't? An answer here would help you to have a strong closing statement.

Figures

Fig1. In the key "chert nodular" should be "chert nodule"

Supplementary File:

Lines 158, 243, 251- the word "criterium" is used, but it should be "criterion". It seems that "criterium" is a bicycle race.

Good job on this manuscript; I hope my comments help.

Sincerely,
Leigh Anne Riedman

Reviewer #2 (Remarks to the Author):

This ms reports on new occurrences of *Horodyskia* from the early Neoproterozoic of North China. *Horodyskia* is problematic at any number of levels - is it a fossil or some sort of sedimentary structure? If it's a fossil, is it prokaryotic or eukaryotic? If it's eukaryotic, is it an alga or something else? This new material is substantially younger than the early Mesoproterozoic type material, and substantially older than specimens reported from the Ediacaran. Its principle novelty is that the conventional 'string-of-beads' bedding plane expression is accompanied by some preserved organic material. This is certainly of passing interest, but it's a bit of a stretch to conflate the presence of some associated organic carbon with "carbonaceous compressions" - a term that implies direct histological/anatomical connectivity to a once-living organism. At least in principle, the organic material associated with the *Horodyskia* beads could derive from localized preservation of (unrelated) microbial material. There are unambiguous carbonaceous compression fossils in these same strata (e.g., *Tawuia*, *Protoarenicola*, *Eosolena*, etc.), but these are decidedly more convincing in terms of continuous morphology-defining films. By contrast, the organic component of *Horodyskia* is incidental to its overall expression.

Even allowing that *Horodyskia* is biogenic, and that organic component in these fossil reflects its original morphology, it has limited implications for resolving its larger-scale affiliations. The authors interpret it as a coenocytic eukaryotic alga based on its size and some general anatomic and taphonomic comparisons with modern organisms. Yes, this is a possible interpretation, but there is nothing compelling or new in the accompanying arguments. Whether or not *Thiomargarita* is a convincing modern analogue is not really the issue here, but whether prokaryotes are capable of attaining macroscopic size in principle. Clearly they can, both as individual cells (the authors seem to have missed the recent Volland et al. paper on coenocytic cm-long *Thiomargarita magnifica*), and as colonies (e.g., *Microcystis*, *Nostoc*). Nor is there reason to assume that the full range of prokaryotic morphology is exhausted by modern diversity: no modern filamentous cyanobacteria exceed 100 μm in diameter, but there is a perfectly sensible case for interpreting mm-diameter *Grypania* as an (extinct) giant cyanobacterium (Sharma & Shukla 2009). In other words, there is nothing diagnostically eukaryotic (or indeed prokaryotic) about the morphology of *Horodyskia*, making it premature to speculate on whether it is photosynthetic or coenocytic or even protistan. If it is eukaryotic, the more topical issue would be whether it represents a crown-group or stem-group eukaryote (Porter 2020) - rather than trying to shoe-horn it into a group of modern algae.

Nick Butterfield

41 – “extremely scarce” is an exaggeration; cf. *Grypania*, *Protoarenicola*, *Tawuia*, etc. ‘relatively scarce’ or ‘recurrent’ would be more accurate.

66 – “mainly preserved as carbonaceous compressions or organic-walled microfossils”. It would be useful to explain exactly what is meant by these terms. Not all carbonaceous residues preserved on bedding surfaces are ‘organic-walled fossils’.

141 – “ The beads are mainly enriched in carbon (Figs. 4h; 5g), consistent with their carbonaceous nature.” Does ‘enriched in carbon’ equal ‘carbonaceous compression’ equal ‘organic-walled microfossil’ (see above comment)? It doesn’t to me.

158 – I agree that the organic C suggests a biological origin, but it’s not immediately obvious why that excludes sedimentary/microbial structures.

186 – if the biogenicity of *Horodyskia* is ‘accepted by most paleontologists’ then it’s not clear what this new ‘organically preserved’ material has to offer. The carbonaceous material does not help to resolve its anatomy or phylogenetic position.

211-218 – it’s not clear how these marginal thickenings and concentric rings relate to the compressional folds observed in flattened spheroidal fossils. They do not look anything like the concentric rings in *Chuarina*, for example. Further explanation/interrogation is required before concluding a vesicular/spheroidal habit for the individual beads.

269 – unclear

277-279 – ‘typical’ prokaryotic cells are tiny, but irrelevant to the present discussion. The fact that “recalcitrant cell walls with a large size are usually considered comparable to modern eukaryotic rather than prokaryotic cells” is simply repeating the current prejudice. The question here is whether large size and/or recalcitrant wall are diagnostically eukaryotic. I don’t think they are, but would be interested to hear any compelling arguments.

284 – this is a laughable conclusion. The fact that extant *Thiomargarita* has less recalcitrant walls than what the authors imagine for *Horodyskia* might conceivably be used to rule out affiliation with sulfur-oxidizing bacteria, but it doesn’t remotely address all prokaryotes that have existed over the past four billion years. One obvious counter-example is *Grypania*, which has been interpreted (not unreasonably) as a giant cyanobacterium.

286 – as above, the large ‘cell size’ of *Horodyskia* is not a diagnostically eukaryotic feature, nor does the absence of preserved cellularity necessarily indicate a coenocytic condition. It’s certainly a possibility, but would need to be accompanied by evidence of well-preserved cell walls more generally (e.g., in associated microfossils). In the absence of such criteria there are no grounds for assuming the individual ‘beads’ were not composites of multiple cells; if the constituent cells lacked secreted cell walls (like animals and most non-photosynthetic protists), then even that test would fail. Without some positive evidence for *Horodyskia* having a coenocytic construction, the extended discussion of various organisms that do or don’t express such a condition carries no weight.

320 – *Horodyskia* certainly could be a coenocytic alga, but it’s simplistic to limit the search to extant groups with such a structure. It would make more sense to start by asking whether it was photosynthetic.

337 – It would be useful to define/distinguish the terms multicellular and coenocytic at some point.

Presumably it's the individual 'beads' of Horodyskia that are being interpreted as coenocytic, whereas the 'string' is multicellular?

362 – Yes, these various macroscopic taxa “may also be eukaryotes” – but the jury is still very much out. This should be more clearly acknowledged in the discussion (e.g., Sharma & Shukla 2009; Butterfield 2015; Porter 2020)

365 – there is nothing in these new Neoproterozoic examples of Horodyskia that adds weight to the argument for their eukaryoticity in the early Mesoproterozoic. At most, the presence of organic carbon supports the argument for biogenicity.

Reviewer #3 (Remarks to the Author):

This manuscript presents well-preserved carbonaceous compressions or organic-walled microfossils of Horodyskia from the Tonian (~950- 720 Ma) Shiwangzhuang Formation in the western Shandong and Jiuliqiao Formation in the Huainan region of North China. Light microscopy (LM), Scanning electron microscope (SEM), Energy dispersive X-ray spectroscopy (EDS) elemental mapping, and Raman spectroscopy are used to ascertain the biogenicity, their taphonomic processes, and affinity to fossils. The fossil material presented in the manuscript is novel and will be of interest to the Precambrian geoscience community because the Horodyskia is described as not more than half a dozen Proterozoic localities in the world.

The authors have done an excellent job describing this fossil material in detail and investigating the preservation of the organisms. All the illustrations in the manuscript are excellent. I think it could be accepted after minor revisions.

Comments:

Line no. 44: site suitable reference.

Line no. 64: reference missing.

Reference no. 29, 35, 36, 37 and 53: incomplete (recheck).

Reviewer comments in black fonts.

Point-to-point responses to reviewers' comments and quotes from revised manuscript are in red fonts, with line numbers (Line xxx) of the revised manuscript cited at the end of each response.

Reviewers' comments:

Reviewer #1 (Remarks to the Author):

Comments on: Tonian carbonaceous compressions shed light on *Horodyskia*, one of the oldest multicellular and coenocytic macro-organisms

The authors present new materials of *Horodyskia moniliformis* and *H. minor* from the Tonian Shiwangzhuang and Jiuliqiao formations and present data based on carbonaceous compressions to suggest a coenocytic eukaryotic body plan for these taxa. *Horodyskia moniliformis* is a well known but enigmatic macrofossil ranging from the early Mesoproterozoic to the end of the Ediacaran; insights to its biological affinity and mode of life are of great interest not only to Precambrian paleontologists, but also evolutionary biologists as both groups grapple with the origin and early evolution of eukaryotes and the pathways to macroscopic size. This paper presents interesting new fossil data and an analysis of body plan and affinity hypotheses. I recommend publication of this manuscript after rather minor revisions and development.

General:

- There are several opportunities to shorten and tighten this manuscript that will benefit the reader and the material. I mention some of these in my line-by-line comments below.

Response: Thanks for the suggestion. We have revised our manuscript as suggested. Please see our responses below.

- There's mention of additional multicellular fossils and reference to comparing Raman data of those and the *Horodyskia* specimens- those data need to be shared, at least in the supplementary files.

Response: Raman data of three multicellular fossils have been added to the Supplementary Raman Data sheets 5, 6 for comparison and cited in the main text (Line 441-446).

- I find myself wanting more morphological analyses of the carbonaceous remains- Do you get anything useful from SEM of the beads? Can you see wrinkles or ripples to the vesicle?

Response: SEM images of *Horodyskia* beads (revised Figs. 4i-k; 5h-l) and representative carbonaceous compression macrofossils (Fig. S2), including *Protoarenicola/Pararenicola*, *Chuarina*, and *Tawuia*, from the Shiwangzhuang Formation have been added. Both the carbonaceous films of *Horodyskia* beads and other macrofossils show typical cracks under SEM. However, unlike organic-walled microfossils, we cannot observe wrinkles from these SEM images, probably because of the fusion of carbonaceous films. Therefore, instead of directly describing the darker structures as “compressional folds”, we described them as thickenings and interpreted them as possible compressional folds (Line 263-265; 490-493).

- Modern testate amoebae are found in several very distant clades, most prominently the Amoebozoa and the Rhizaria. Rhizarian testate amoebae do not have a well documented deep fossil record; almost all fossil testate amoebae belong to the Amoebozoa. I have pointed out changes that need to be made to accommodate this. Response: We have revised our manuscript as suggested (Line 543-547; 560-561). Please see our responses below.

- Although I think it’s technically ok since there are not genera or species erected here, having nomenclatural acts such as emendations in the supplementary files should be avoided.

Response: We have moved the “Systematic Paleontology” part from the supplementary file to the “Results” section of the main text (Line 133-428).

Line-by-line:

Lines 21-22- You can delete these first two sentences for a stronger start. It isn’t necessary to explain why macroscopic fossils are important and the first sentence doesn’t actually convey any information. The second sentence has a few problems- what do you mean by “unambiguous macrofossils”? It isn’t that their macro- status ambiguous. I’d guess you mean that their affinity is ambiguous- but that isn’t clear here. Secondly, the phrase “earliest crown-group eukaryotes” needs conditions placed on it- those around 1 Ga are the earliest fossils that have been allied in the literature with crown groups, but two things: 1) if those fossils are representatives of crown groups- we can’t say that earlier crown group eukaryotes didn’t exist- certainly they must have, in fact; and 2) those are interpretations that those ~1Ga fossils are crown-group and even if that interpretation is generally accepted, it is still an interpretation and that needs to be stated.

However, the larger point here is that you could cut those first two sentences and get right to the point with talking about *Horodyskia*. E.g. start with “*Horodyskia* is one of the few...” and you might consider combining it with the next sentence with a “but the biological interpretation...”

Response: The first sentence has been removed as suggested. However, we think that it is necessary to keep the second sentence to introduce macrofossils predating the Mesoproterozoic-Neoproterozoic transition. We have revised the second sentence as suggested (Line 24-26).

Line 27- purely stylistic, but if you chose to say “has been”, you should give a since of time, like “the biological interpretation of this enigmatic fossil, however, has been a matter of controversy since its initial description in 2000”. Otherwise you might change ‘has been’ to ‘is’. Incidentally, in looking to remind myself of when *Horodyskia* was first described, I noticed that you don’t include Yochelson and Fedonkin (2000) in your references. If you have space for its inclusion I very much recommend you include it.

Response: We have added “since its discovery 1982” to the sentence as suggested (Line 29). Yochelson and Fedonkin (2000) has been added where *Horodyskia* is first mentioned in the Introduction section (Line 51-53).

Lines 27-30- Since the beginning of this abstract set up the fact that there is controversy about the biological interpretation of *Horodyskia*, the reader expects the next few sentences to indicate that you will address that matter with your materials.

Response: The following two sentences have been revised to highlight the biogenicity and biological affinity of *Horodyskia* reinforced by our materials with organic walls (Line 30-35).

Line 30- can you be more explicit about what you mean by “precious link”?

Response: Two species of *Horodyskia*, *H. moniliformis* and *H. minor*, with distinct bead sizes are closely preserved on the same bedding plane. *H. moniliformis* is typically preserved in Mesoproterozoic successions and *H. minor* is typically preserved in Ediacaran successions. Whether *H. minor* should be assigned to *Horodyskia* is highly debated in the last decade. Our Tonian materials therefore provide a precious link, both temporally and biologically, between the early Mesoproterozoic (*H. moniliformis*) and the Ediacaran (*H. minor*) horodyskids. This sentence has been revised accordingly (Line 35-38).

Lines 31-34- I’m not sure what you’re wanting to do with this sentence- basically it is saying that *Horodyskia* is interpreted as a coenocytic eukaryote and it is old and long ranging. This thought needs to be more developed here- what about the existence of coenocytic eukaryotes at 1.5 Ga will help with our understanding of eukaryotic evolution or the evolution of a multicellular form? Additionally, the sentence structure is somewhat difficult to parse- is there a reason we would exclude or not take into consideration the specimens from Montana (the type specimens)?

Response: The ending sentence has been re-written, to highlight that this study has provides evidence that some eukaryotes have acquired macroscopic size through the combination of coenocytism and simple clonal coloniality at least ~1.48 Ga, and also

highlights an exceptionally long range and morphological stasis of this Proterozoic genus of macrofossils (Line 38-42).

Line 40- It's generally not acceptable in formal writing to use "but" as the first word in a sentence. You could side-step this by instead using "However".

Response: We have replace "But" with "However" as suggested (Line 48-51).

Line 43: you might add the word "evolution" at the end of this sentence. It's not really the tempos and modes of individual early eukaryotes that we're interested in here, but the tempo and mode of early eukaryotic evolution.

Response: The word "evolution" has been added as suggested (Line 48-51).

Line 45- you might consider switching the ages here to move from older to younger. It makes more sense that the stratigraphic range extends from the early Mesoproterozoic to the terminal Ediacaran.

Response: Changes have been made accordingly to this place and elsewhere as suggested (Line 26-28; 51-53; 56-59)

Line 45- Rather than "It is", you might say "This genus is" to be more precise in wording.

Response: "It is" has been replaced by "This genus is" as suggested (Line 53-56).

Line 51- I appreciate having more insight on the likely age of the Backdoor and Stag Arrow fms, but you need to include a reference for why the age is probably 1.21-1.07 Ga rather than the other age range you state.

Response: The ages of the Backdoor and Stag Arrow formations of the Bangemall Supergroup in Western Australia and the Balfour Subgroup of the Rocky Cape Group in Tasmania have been updated. Two new references (Cutten et al., 2021; Halpin, et al., 2014) about these ages have been added (Line 59-63).

References:

Cutten, H., Johnson, S., Zwingmann, H., Todd, A. & Uysal, T., 2021, Dating Proterozoic Fault Movement Using K-Ar Geochronology of Illite Separated from Lithified Fault Gouge: Geological Survey of Western Australia. v. Report 214, p. 1–25.

Halpin, J. A. et al., 2014, Authigenic monazite and detrital zircon dating from the Proterozoic Rocky Cape Group, Tasmania: Links to the Belt-Purcell Supergroup, North America: Precambrian Research, v. 250, p. 50–67.

Line 56- there's no reason for the wide global distribution and long stratigraphic range to inform on the biological affinity of Horodyskia, so the "in spite of" is a little odd here. You might get the same idea across, i.e., that even though we have lots of specimens to study, we still don't know who Horodyskia is, by simply saying, "Horodyskia has a wide global distribution and long stratigraphic range but its

biological affinity remains controversial...” or if you really want to keep “in spite of”, you should make the point that its in spite of the many specimens to study, not in spite of the stratigraphic range and global distribution.

Response: This sentence has been revised as suggested (Line 65-66).

Line 57-58- I like the phrase, “even the debate on its biogenicity swings back and forth.”

Response: Thanks.

Line 59- it seems that pseudofossil and dubiofossil should be mentioned back-to-back here.

Response: These two words have been placed together as suggested (Line 66-70).

Lines 62-70- This! This needs to be mentioned in the abstract- boil down to one sentence and that can be somewhere near the end, followed by quick results. It is a big deal that you’re reporting organic-walled specimens of *Horodyskia* and that is something that sets this work apart from previous attempts to discern *Horodyskia*’s affinity.

Response: Thanks for the suggestion. The abstract has been revised accordingly to stress that previous reported specimens of *Horodyskia* do not preserve organic walls and that our materials have organic walls which therefore can reinforce the biogenicity of *Horodyskia* (Line 30-35; 86-89).

Lines 73-77- what do you mean by “constraints”- age constraints? This wording is vague and needs some attention. What precisely do these fossils offer in terms of a better understanding of origins of multicellularity, the early evolution of multicellularity or the origin or early evolution of the coenocytic body plan. And what does knowing there were multicellular organisms or coenocytes in the Mesoproterozoic tell us? Try to be specific about what your fossils can tell us (and tell us what they can’t say!) rather than using vague and imprecise terms.

This sentence makes a bit more sense here than it does in the abstract because it has more context, but the problem remains as to why one would need to exclude the material from Montana- 1) why those fossils? Because they’re very old and there’s a concern about the affinity interpretation being extended to 1.48 Ga? 1a) then that means you’re questioning the identification of your materials as *Horodyskia*, as the fossils from Montana are the type materials and standard bearers. 1b) if it is about the age, why not be concerned about extending the affinity to fossils of Western Australia and Tasmania that are also Mesoproterozoic in age? 2) why would the age matter? If those are all really the same biological entity, i.e., all specimens you’ve mentioned in lines 48-55 are *Horodyskia* and it is monophyletic, then they should all have the same biological affinity. So this is a strange sentence- if there’s a reason to consider the Montana fossils as separate, that needs to be discussed and the taxonomic situation needs to be sorted. If you’re just trying to make the point that a coenocytic body plan dates to at least as old as 1.48 Ga, then say that.

Response: Thanks for this persuasive suggestion. This ending sentence has been divided into three sentences to be more accurate and specific (Line 84-92).

Line 82- I would say it's not necessary to say "morpho- species". It's understood that the species are distinguished by morphological characters and there is the general assumption about biological relatedness applied to all fossil species, so unless there is good reason to think that Horodyksia is polyphyletic, then you could just say "species".

Response: Corrected (Line 98-102).

Line 85- I find the idea of having nomenclatural acts, including emendation of genera and species, in supplementary files very worrying. I think as long as the supplementary files are hosted as pdfs with their own ISSN or ISBN, then they may be acceptable as "effective publication" according to the International Code of Nomenclature for Algae, Fungi and Plants- and I guess since you're not naming new genera or species here it won't cause a problem, but I'd expect that the emendation will not get the notice and application it otherwise would get if published in main text. For reference, here is a link to one of the key sections on effective publication.

https://www.iapt-taxon.org/nomen/pages/main/art_29.html

Response: Thanks for this suggestion. We have moved the "Systematic Paleontology" part from the supplementary file to the "Results" section of the main text (Line 133-428).

Lines 114-115- The phrase "which also occur in aggregation" is out of place here- it seems like you're saying that the Jiuliqiao Formation also occurs in aggregation.

Response: The paragraph has been removed since the "Systematic Paleontology" part has been added here.

Line 147- The chemospace discussion needs more of a set up before you discuss results- just one additional sentence should suffice.

Response: A sentence has been added to introduce chemospace (Line 126-127).

Line 161- please say more about the coexisting multicellular fossils- what taxa? Have they been published? If so, give the reference; if not maybe put them in supplementary files. –Ah, you do mention them. Maybe for the reader's benefit put a parenthetical here saying something like (briefly discussed below)

Response: The three analyzed specimens of multicellular fossils include a cell tetrad, a multi-chained cell colony, and a spheroidal cell colony. Their information has been added to the Supplementary Raman Data sheets 5 and 6.

Line 164- If you're going to refer to data from those multicellular fossils, you need to include those data and information about the multicellular fossils.

Response: The information about the taxonomic and Raman data of these multicellular fossils has been added to the Supplementary Raman Data sheets 5 and 6.

Line 168-169- I'm not sure why you're repeating the shared thermal history point here. It's already been said and this sentence is otherwise about interpretations of the organic walls, which needs a bit more development.

Response: This phrase has been removed and the sentence has been revised to stress the beads are bona fide fossil vesicles (Line 446-450).

Line 189-191- If this is the first co-occurrence of these two species, that should also be mentioned in the abstract when you mention that it is the first occurrence of organic-walled preservation.

Response: The information about the first co-occurrence of these two species has been added to the abstract (Line 35-38).

Line 203- I think I might put in a "likely" here. I.e., "... suggesting that they are not only biogenic but also likely congeneric organisms". That is not a critical recommendation, but I, personally, would put it in.

Response: Added (Line 479-483).

Line 204- if these taxa were all synonymized, please cite reference.

Response: These two synonyms in the parentheses have been removed here since the "Systematic Paleontology" part has been added to the main text.

Line 295- this seems to suggest that arcellinid testate amoebae are rhizarians, which is not accurate. Arcellinids are within the Amoebozoa and rhizarians are a branch within SAR; they are quite separate clades. Perhaps this is not what you meant and its just unfortunate sentence construction- this can also be read to say that there are coenocytic algal groups within the Rhizaria- I am not an expert on modern rhizarian diversity, but I am unaware of coenocytic photosynthesizing rhizarians so it might benefit your readers for you to both reword this sentence and to also expand it to give more details on these living groups.

Response: Corrected. This phrase has been revised to "including arcellinid testate amoebae, foraminifers, and some algal groups" (Line 543-547).

Line 298- regarding the references- it is true that rhizarians can make organic tests and Adl et al do mention that, but the paper by Porter and Riedman is only about arcellinid testate amoebae, which within the Amoebozoa, a completely separate group from the Rhizaria. Additionally, a quick look at the cited work of Kutluk and Mazei shows fossil arcellinids (Amoebozoa), not rhizarian testate amoebae.

Response: The phrase "rhizarian amoebae" has been corrected as "arcellinid testate amoebae" here (Line 548-549).

Lines 297-300- the idea of Horodyskia as testate amoebae has never made sense to me- the testate amoebae (Luo and Miao focus on those shaped like Arcella), but these testate amoebae are so small- like you mention in your paper, they are 50 to 200

microns wide (although at least the fossil testate amoebae tend to be much smaller than 200 μm , often less than 100 μm)- they are so small- an order of magnitude smaller than *Horodyskia moniliformis* beads and a half to less than a third of the diameter of *H. minor*. Aside from the size problem, testate amoebae, fossil or modern, don't form strings. Some *Horodyskia* have stolons connecting them- what would that be if these were testate amoebae? And yes, the lack of an aperture is an issue, but Luo and Miao focused on Arcella-type testate amoebae, so the aperture wouldn't be prominent. It is just such a strange argument, it seems like it derails your paper to present this hypothesis so prominently. Obviously you have to address this hypothesis, but what might help is to signal to the reader that in the following paragraphs you will be evaluating recent hypotheses of affinity- maybe with a subheading or just a sentence or clause before launching into it.

Response: This paragraph has been divided into two new paragraphs, to discuss separately the affinity of *Horodyskia* as testate amoebae proposed by Luo and Miao, 2020 and as foraminifers proposed by Dong et al., 2008 (Line 548-577).

Line 302- This sentence needs to be changed to accommodate the fact that the testate amoebae discussed in the preceding sentence are not rhizarians.

Response: Corrected (Line 548-549).

Line 308-319- I don't have the answer for this, but in reading this passage I wonder what the occurrence of say, monothalamids, in the Mesoproterozoic would require in terms of other crown groups that would have had to appear before 1.48 Ga. How derived are these groups you mentioned? By 1.48 Ga we would have to have already evolved (that we have ?no? fossil evidence of?)

Response: It is interesting to note that Monothalamids are the earliest foraminifera group (but likely a paragroup) divergent from all the foraminifera groups, including *Tubothalamea* and *Globothalamea* (Pawlowski et al., 2013, *Marine Micropaleontology* 100: 1–10). Therefore, at least it does not require other crown groups appear before 1.48 Ga.

Line 318- It occurs to me that we probably need to use a skeptical eye when considering some molecular clock dates that use *Bangiomorpha* as a calibration point, but that predate its more tightly constrained age reported by Gibson et al (2017) that is ~200 million years younger than previously reported. And I would be particularly concerned about the older age reported by Pawlowski et al (2003) since in that paper they also used as a calibration point, some VSMs from the Chuar Group (Porter et al, 2003) that were even then only tentatively allied with rhizarian testate amoebae (*Imbricatea*). Since publication of that VSM paper in 2003, that once tentative suggestion of Rhizaria in the Tonian has not been supported and is thought to be less likely for various reasons. The point of this is that those ages for the emergence of the Foraminiferan Phylum are likely too old based on calibrations we now consider wrong.

Response: The reference Pawlowski et al. (2003) has been removed from the discussion (Line 555-559).

Line 330-336- How do these algae reproduce? Is there any part of their lifecycle that could look like a dividing *Horodyskia* bead? Do you find multiple bubbles of *Valonia* or other algae in association with each other, if not actually connected and in a ‘string’?

Response: The reproduction of *Valonia* is achieved by segregative cell division and lenticular cell division, the latter of which looks more like a dividing *Horodyskia* bead (Olsen and West, 1988, *Phycologia*, 27:103-108). The cells/bubbles of *Valonia* are usually occur in close aggregation with each other and sometimes a few of the cells are found in a short string (see picture below).

Sources: https://www.redseadiving.cz/Data/Sites/1/media/valonia_ventricosa2.jpg

Line 335- More precisely, these specific modern, extant, coenocytic algae are imperfect matches for *Horodyskia*. It might be worth making the point that we don't have to find an exact match, do we really expect those species to persist unchanged for 1.5 billion years?

Response: We have added this point of view to the following sentence (Line 594-598).

Line 347-351- This is an unnecessarily complex sentence. It seems like you're trying to say that the new data collected from the Shiwangzhuang specimens suggests a coenocytic nature for *Horodyskia*, a fossil genus that dates from the early Mesoproterozoic. I like that you're then bringing in to the discussion some of the other macrofossils of the early Proterozoic; it is worth considering if they also might have this body plan- what should one look for in those taxa that would suggest this?

Response: This complex sentence has been shortened (Line 608-610). The proposal that we should evaluate whether other Mesoproterozoic macrofossils also possess a coenocytic body plan has been added to the end of this paragraph (Line 614-617).

Line 360-361- These taxa are known from more units than just the Ruyang and Roper, so it's a little odd to mention only those two. You could just cut the unit names (but keep the citations) since your point is about time, not the specific units.

Response: Corrected (Line 618-621).

Line 364- This is the same point I've made before, but Why would you exclude the Appekunny Fm? I don't understand the point of including this sentence. If you're just trying to hedge your bets, don't. Just be full-throated about saying that your data suggest that *Horodyskia* was a macroscopic coenobium, so the macroscopic coenocytic body plan was present in the Mesoproterozoic (and probably since then- I say probably because that body plan likely has evolved repeatedly).

Response: This sentence has been revised as suggested (Line 623-627).

Line 383- be specific about what the "valuable link" is- is it that you're seeing the two taxa co-occurring?

Response: This sentence has been revised as suggested (Line 644-647).

Line 382- you mention detailed characterization of the carbonaceous compressions, and you do give chemical data, but can you give more morphological data? Can you see evidence of wrinkles or ripples in the vesicle if you look with SEM?

Response: More morphological data, including light microscopy and scanning electron microscopic images, have been added as suggested (revised Figures 3g, h; 4i-k; 5h-l; Supplementary Figure 2). Cracks are the main characteristics observed under SEM for the carbonaceous compressions of *Horodyskia* and other representative carbonaceous compression macrofossils. Wrinkles or ripples cannot not been observed, therefore, we have changed the description to "thickenings" and interpreted them as possible compressional folds (Line 263-265, 490-493).

Line 387-392- This last sentence can be made much shorter and to-the-point. What you might do is to cut the beginning about the Appekunny and start with "Our study reports evidence that *Horodyskia*, a genus of macroscopic fossils dating from the early Mesoproterozoic to the terminal Ediacaran, attained its macroscopic size through the combination of coenocytism and simple clonal coloniality. This fossil and

its body plan provide an important constraint for the origin and early evolution of coenocytic eukaryotes.”

Response: Thanks for suggestion for these ending sentences. We have revised the last sentence as suggested (Line 650-655).

Regarding: Simple clonal coloniality. This wasn't discussed, so I can't see how you can argue that simple clonal coloniality contributed to Horodyskia's large size. Also, I'm not sure coloniality is a word.

Response: “Coloniality” is a word to classify the degrade of multicellularity (Butterfield, 2009, Precambrian Research 173: 201–211). Butterfield (2009) has adopted a similar term "simple cellular coloniality" in his paper of “Modes of pre-Ediacaran multicellularity”. An interpretation for this term has been added to the paragraph above (Line 601-605).

Regarding: constraint for the origin and early evolution of coenocytic eukaryotes. This is still a vague statement and repeating it just makes the reader more aware of how vague it is. What is being constrained? Timing? What about early eukaryotic evolution can be better understood in light of these new data? Is there something inherent in this body plan that will allow an organism to do something it otherwise couldn't? to exploit a niche/live somewhere otherwise unavailable? Would a coenocytic organism have an effect on other organisms or the environment around it that some other style of macro-body plan wouldn't? An answer here would help you to have a strong closing statement.

Response: The ending sentence has been revised to suggest that the time of origin of coenocytic eukaryotes might be constrained (Line 650-655).

Figures

Fig1. In the key “chert nodular” should be “chert nodule”

Response: Corrected.

Supplementary File:

Lines 158, 243, 251- the word “criterium” is used, but it should be “criterion”. It seems that “criterium” is a bicycle race.

Response: Corrected (Line 221, 290, 298 in the main text).

Good job on this manuscript; I hope my comments help.

Sincerely,

Leigh Anne Riedman

Reviewer #2 (Remarks to the Author):

This ms reports on new occurrences of Horodyskia from the early Neoproterozoic of North China. Horodyskia is problematic at any number of levels - is it a fossil or some sort of sedimentary structure? If it's a fossil, is it prokaryotic or eukaryotic? If it's

eukaryotic, is it an alga or something else? This new material is substantially younger than the early Mesoproterozoic type material, and substantially older than specimens reported from the Ediacaran. Its principle novelty is that the conventional 'string-of-beads' bedding plane expression is accompanied by some preserved organic material. This is certainly of passing interest, but it's a bit of a stretch to conflate the presence of some associated organic carbon with "carbonaceous compressions" – a term that implies direct histological/anatomical connectivity to a once-living organism. At least in principle, the organic material associated with the *Horodyskia* beads could derive from localized preservation of (unrelated) microbial material. There are unambiguous carbonaceous compression fossils in these same strata (e.g., *Tawuia*, *Protoarenicola*, *Eosolena*, etc.), but these are decidedly more convincing in terms of continuous morphology-defining films. By contrast, the organic component of *Horodyskia* is incidental to its overall expression.

Response: In the revised manuscript, we have added new data of SEM images of the *Horodyskia* specimens and other unambiguous carbonaceous compression macrofossils, including *Protoarenicola/Pararenicola*, *Chuarua*, and *Tawuia*, from the Shiwangzhuang Formation (revised Figures 4i-k; 5h-l; Supplementary Figure 2). The carbonaceous nature of the *Horodyskia* specimens is quite similar to those of the carbonaceous compression macrofossils. They are mainly composed of organic carbon and show characteristic cracks which are typically observed in Tonian carbonaceous compression fossils (e.g., Figure 2I, J in Tang et al., 2021, *Gondwana Research* 97: 22–33). In those three-dimensionally preserved specimens (Figure 2I-p), the organic-walls of the fossil are well-preserved, indicating these are organic-walled fossils. Raman spectroscopy also shows that the carbonaceous material of the *Horodyskia* specimens have spectral characteristics similar to co-existing multicellular fossils, indicating that they both experienced low-grade metamorphism with apparent peak metamorphic temperatures (Supplementary Raman Data sheets 1, 5, 6). As far as we know, there are no microbial material that can present a similar string of beads morphology, bimodal size distribution with a narrow average/standard deviation ratio, and regular bead spacing of *Horodyskia*. The most parsimonious interpretation is that *Horodyskia* is a bona fide fossil which preserves the organic remains of an ancient organism.

Even allowing that *Horodyskia* is biogenic, and that organic component in these fossil reflects its original morphology, it has limited implications for resolving its larger-scale affiliations. The authors interpret it as a coenocytic eukaryotic alga based on its size and some general anatomic and taphonomic comparisons with modern organisms. Yes, this is a possible interpretation, but there is nothing compelling or new in the accompanying arguments. Whether or not *Thiomargarita* is a convincing modern analogue is not really the issue here, but whether prokaryotes are capable of attaining macroscopic size in principle. Clearly they can, both as individual cells (the authors seem to have missed the recent Volland et al. paper on coenocytic cm-long *Thiomargarita magnifica*), and as colonies (e.g., *Microcystis*, *Nostoc*). Nor is there reason to assume that the full range of prokaryotic morphology is exhausted by

modern diversity: no modern filamentous cyanobacteria exceed 100 μm in diameter, but there is a perfectly sensible case for interpreting mm-diameter *Grypania* as an (extinct) giant cyanobacterium (Sharma & Shukla 2009). In other words, there is nothing diagnostically eukaryotic (or indeed prokaryotic) about the morphology of *Horodyskia*, making it premature to speculate on whether it is photosynthetic or coenocytic or even protistan. If it is eukaryotic, the more topical issue would be whether it represents a crown-group or stem-group eukaryote (Porter 2020) – rather than trying to shoe-horn it into a group of modern algae.

Nick Butterfield

Response: The reference Volland et al. (2022) has been cited in the revised manuscript (Line 526-534). Although *Ca. Thiomargarita magnifica* can reach a centimetric length, it has a long tubular morphology and its cell volumes are between $2.37 \times 10^5 \mu\text{m}^3$ and $2.20 \times 10^7 \mu\text{m}^3$ (Table S2 in Volland et al., 2022), which are an order of magnitude less than the largest volumes ($2.2 \times 10^8 \mu\text{m}^3$) of those spherical cells of *Thiomargarita namibiensis* (Table S2 in Pang et al., 2018). So the recent finding of *Ca. Thiomargarita magnifica* does not break through our previous understanding of the capability of attaining a large size for prokaryotes. Nonetheless, both the volumes of *Thiomargarita namibiensis* and *Ca. Thiomargarita magnifica* are at least two orders of magnitude less than the largest volumes ($2.3 \times 10^{10} \mu\text{m}^3$) of the beads of *Horodyskia*. Colonies of prokaryotes do not present binary division or bi-modal size distribution with a narrow average/standard deviation ratio. Therefore, we do not think that individual prokaryotic cells or prokaryotic colonies could be good analogs for *Horodyskia*. The Proterozoic macrofossil *Grypania/Katnia* has been interpreted by one group of researchers as giant cyanobacterium (Sharma & Shukla 2009). But this cyanobacterial interpretation has not been widely accepted. Most of the Precambrian paleontologists regarded this taxon as macroalga or eukaryote (Butterfield, 2009; Han & Runnegar, 1992; Hofmann, 1985, 1992; Javaux and Lepot, 2018; Knoll et al., 2006; Niu, 1998; Runnegar, 1991; Sun et al., 2006; Walter et al., 1976, 1990; Wang et al., 2016; Xiao, 2013). Anyway, *Grypania* cannot be used as an unambiguous cyanobacterial fossil here which would cause circularity to this argument. However, the cell volume of *Grypania/Katnia* ($\leq 5.2 \times 10^8 \mu\text{m}^3$; Table S2 in Pang et al., 2018) is still two orders of magnitude less than the volume ($\leq 2.3 \times 10^{10} \mu\text{m}^3$) of the beads of *Horodyskia*. The interpretation that *Grypania/Katnia* is a giant cyanobacterium, even if true, would not falsify the interpretation that *Horodyskia* is a eukaryote. In the revised manuscript, we have interpreted *Horodyskia* as total-group eukaryote, since we do not have more characteristics to determine whether it is a crown-group or stem-group eukaryote (Line 594-598).

References:

- Butterfield, N. J., 2009, Modes of pre-Ediacaran multicellularity: Precambrian Research, v. 173, p. 201–211.
- Han, T. M., and Runnegar, B., 1992, Megascopic eukaryotic algae from the 2.1-billion-year-old Negaunee Iron-Formation, Michigan: Science, v. 257, p. 232–235.

- Hofmann, H. J., 1985, The mid-Proterozoic Little Dal macrobiota, Mackenzie Mountains, north-west Canada: *Palaeontology*, v. 28, no. 2, p. 331–354.
- Hofmann, H. J., 1992, Proterozoic carbonaceous films, in Schopf, J. W., and Klein, C., eds., *The Proterozoic Biosphere: A Multidisciplinary Study*: New York, Cambridge University Press, p. 349–357.
- Javaux, E. J., and Lepot, K., 2018, The Paleoproterozoic fossil record: Implications for the evolution of the biosphere during Earth's middle-age: *Earth-Science Reviews*, v. 176, no. Supplement C, p. 68–86.
- Knoll, A. H., Javaux, E. J., Hewitt, D., and Cohen, P., 2006, Eukaryotic organisms in Proterozoic oceans: *Philosophical Transactions of the Royal Society B: Biological Sciences*, v. 361, p. 1023–1038.
- Niu, S., 1998, Confirmation of the genus *Grypania* (megascopic alga) in Gaoyuzhuang Formation (1434 Ma) in Jixian (Tianjin) and its significance: *Progress in Precambrian Research*, v. 21, p. 38–46.
- Pang, K., Tang, Q., Chen, L., Wan, B., Niu, C., Yuan, X., and Xiao, S., 2018, Nitrogen-fixing heterocystous cyanobacteria in the Tonian Period: *Current Biology*, v. 28, no. 4, p. 616–622.
- Runnegar, B., 1991, Precambrian oxygen levels estimated from the biochemistry and physiology of early eukaryotes: *Palaeogeography, Palaeoclimatology, Palaeoecology*, v. 5, no. 1-2, p. 97–111.
- Sharma, M., and Shukla, Y., 2009, Taxonomy and affinity of Early Mesoproterozoic megascopic helically coiled and related fossils from the Rohtas Formation, the Vindhyan Supergroup, India: *Precambrian Research*, v. 173, p. 105–122.
- Sun, S., Zhu, S., and Huang, X., 2006, Discovery of megafossils from the Mesoproterozoic Gaoyuzhuang Formation in the Jixian section, Tianjin and its stratigraphic significance: *Acta Palaeontologica Sinica*, v. 45, p. 207–220.
- Volland, J.-M., Gonzalez-Rizzo, S., Gros, O., Tylm, T., Ivanova, N., Schulz, F., Goudeau, D., Elisabeth, N. H., Nath, N., Udworthy, D., Malmstrom, R. R., Guidi-Rontani, C., Bolte-Kluge, S., Davies, K. M., Jean, M. R., Mansot, J.-L., Mouncey, N. J., Angert, E. R., Woyke, T., and Date, S. V., 2022, A centimeter-long bacterium with DNA contained in metabolically active, membrane-bound organelles: *Science*, v. 376, p. 1453–1458.
- Walter, M. R., Oehler, J. H., and Oehler, D. Z., 1976, Megascopic algae 1300 million years old from the Belt Supergroup, Montana: A reinterpretation of Walcott's *Helminthoidichnites*: *Journal of Paleontology*, v. 50, no. 5, p. 872–881.
- Walter, M. R., Du, R. L., and Horodyski, R. J., 1990, Coiled Carbonaceous Megafossils from the Middle Proterozoic of Jixian (Tianjin) and Montana: *American Journal of Science*, v. 290A, p. 133–148.
- Wang, Y., Wang, Y., and Du, W., 2016, The long-ranging macroalga *Grypania spiralis* from the Ediacaran Doushantuo Formation, Guizhou, South China: *Alcheringa: An Australasian Journal of Palaeontology*, v. 40, no. 3, p. 1–10.
- Xiao, S., 2013, Written in stone: the fossil record of early eukaryotes, in Trueba, G., and Montúfar, C., eds., *Evolution from the Galapagos: Two Centuries After Darwin*, Volume 2: New York, Springer, p. 107–124.

41 – “extremely scarce” is an exaggeration; cf. *Grypania*, *Protoarenicola*, *Tawuia*, etc. ‘relatively scarce’ or ‘recurrent’ would be more accurate.

Response: The phrase “extremely scarce” has been replaced by “relatively scarce” as suggested (Line 48-51).

66 – “mainly preserved as carbonaceous compressions or organic-walled microfossils”. It would be useful to explain exactly what is meant by these terms. Not all carbonaceous residues preserved on bedding surfaces are ‘organic-walled fossils’.

Response: This sentence has been revised to stress the diverse preservational styles, including carbonaceous compressions, shallow impressions, three-dimensional organic-walls, and casts and molds, of our new material from North China (Line 78-81). Please also refer to our response above in terms of the biogenicity of these microfossils.

141 – “ The beads are mainly enriched in carbon (Figs. 4h; 5g), consistent with their carbonaceous nature.” Does ‘enriched in carbon’ equal ‘carbonaceous compression’ equal ‘organic-walled microfossil’ (see above comment)? It doesn’t to me.

Response: The carbonaceous film of *Horodyskia* specimens show cracks (revised Figures 4i-k; 5h-l) similar to those typically observed in Tonian carbonaceous compressions fossils (Supplementary Figure 2) and probably caused by shrinkage of organic-walls during late diagenetic geopolymerization, kerogenization, and carbonization processes. In those three-dimensionally preserved specimens (Figure 2l-p), the organic-walls of the fossil are well-preserved. Therefore, we think that the presence of carbon in the specimens is consistent with their biogenic and carbonaceous nature. Please also refer to our response above in terms of the biogenicity of these microfossils.

158 – I agree that the organic C suggests a biological origin, but it’s not immediately obvious why that excludes sedimentary/microbial structures.

Response: Organic carbon, three-dimensionally preserved organic-walls with a consistent morphology, and regular bead size and spacing together suggests a biological origin. This sentence has been revised to stress that the carbonaceous nature of these specimens are similar to those representative carbonaceous compression microfossils from the Shiwangzhuang Formation, and that the carbonaceous compression specimens and three-dimensionally preserved organic-walled specimens with a consistent morphology and regular size exclude the possibilities of sedimentary structures (Line 434-450). The possibility of microbial structures have already been discussed in the following paragraph (Line 451-466). As far as we know, there are not any sedimentary or microbial structures that show a bi-modal size distribution with a narrow average/standard deviation ratio (Rouillard et al., 2018, *Geobiology*, 16: 1–18).

186 – if the biogenicity of *Horodyskia* is ‘accepted by most paleontologists’ then it’s not clear what this new ‘organically preserved’ material has to offer. The carbonaceous material does not help to resolve its anatomy or phylogenetic position.
Response: The phrase has been revised to “While its biogenicity has been in dispute over decades” (Line 467). The three-dimensional organic-walled beads and non-organic halos in the North China material do help us to better understand the anatomy of *Horodyskia*, which in turn help us test different phylogenetic interpretations for this taxon.

211-218 – it’s not clear how these marginal thickenings and concentric rings relate to the compressional folds observed in flattened spheroidal fossils. They do not look anything like the concentric rings in *Chuar*, for example. Further explanation/interrogation is required before concluding a vesicular/spheroidal habit for the individual beads.

Response: SEM images of *Horodyskia* beads (revised Figs. 4i–k; 5h–l) have been added. Both the carbonaceous films of *Horodyskia* beads show typical cracks under SEM. However, unlike organic-walled microfossils, we cannot observe wrinkles from these SEM images, probably caused by fusion of carbonaceous films. Therefore, instead of directly describing the darker structures as “compressional folds”, we described them as marginal thickenings and interpreted them as possible compressional folds (Line 263-265; 490-493).

269 – unclear

Response: These two sentences have been revised to be more clear (Line 517-520): “Grey et al.²⁵ had a critical review on all the phylogenetic interpretations of *Horodyskia* proposed prior to 2010 by paleontologists. However, the authors in Grey et al.²⁵ failed to reach a consensus, although they favored the seaweed and hydrozoan interpretations.”

277-279 – ‘typical’ prokaryotic cells are tiny, but irrelevant to the present discussion. The fact that “recalcitrant cell walls with a large size are usually considered comparable to modern eukaryotic rather than prokaryotic cells” is simply repeating the current prejudice. The question here is whether large size and/or recalcitrant wall are diagnostically eukaryotic. I don’t think they are, but would be interested to hear any compelling arguments.

Response: Please see our responses above and below. As far as we know, the largest cells of extant prokaryotes ($\leq 2.2 \times 10^8 \mu\text{m}^3$ for *Thiomargarita*; Table S2 in Pang et al., 2018) and the largest cells of possible cyanobacterial fossils ($\leq 4.1 \times 10^7 \mu\text{m}^3$ for *Anhuithrix* and $\leq 5.2 \times 10^8 \mu\text{m}^3$ for *Grypania/Katnia*; Table S2 in Pang et al., 2018) are at least two orders of magnitude smaller than the beads ($\leq 2.3 \times 10^{10} \mu\text{m}^3$) of *Horodyskia* (Line 531-534). We do think that such an extent of cell size ($10^{10} \mu\text{m}^3$) can be used as a diagnostic characteristic for eukaryotes.

284 – this is a laughable conclusion. The fact that extant *Thiomargarita* has less recalcitrant walls than what the authors imagine for *Horodyskia* might conceivably be used to rule out affiliation with sulfur-oxidizing bacteria, but it doesn't remotely address all prokaryotes that have existed over the past four billion years. One obvious counter-example is *Grypania*, which has been interpreted (not unreasonably) as a giant cyanobacterium.

Response: Since in prokaryotes, only sulfur bacteria can reach a millimetric cell size, we mainly focused our discussion on sulfur bacteria in this paragraph (Line 522-536). We do not know whether there are other groups, including extinct ones, of prokaryotes can reach a millimetric cell size. The fundamental limitation lies in genetics and metabolism between bacteria and eukaryotes (Demoulin and Janssen, 1981, Br. Phycol. J 16: 55–58). It seems impossible for a prokaryotic cell to reach the same size as the beads of *Horodyskia* ($\leq 2.3 \times 10^{10} \mu\text{m}^3$). As we mentioned above, whether the Proterozoic macrofossil *Grypania/Katnia* is a giant cyanobacterium (Sharma & Shukla 2009) or a eukaryotic macroalga (Butterfield, 2009; Han & Runnegar, 1992; Hofmann, 1985, 1992; Javaux and Lepot, 2018; Knoll et al., 2006; Niu, 1998; Runnegar, 1991; Sun et al., 2006; Walter et al., 1976, 1990; Wang et al., 2016; Xiao, 2013) is not settled down. Nevertheless, the cell size of *Grypania/Katnia* ($\leq 5.2 \times 10^8 \mu\text{m}^3$; Table S2 in Pang et al., 2018) is still two orders of magnitude less than the size ($\leq 2.3 \times 10^{10} \mu\text{m}^3$) of the beads of *Horodyskia*. *Grypania/Katnia* would not be a counter-example even if it is a giant cyanobacterium.

286 – as above, the large 'cell size' of *Horodyskia* is not a diagnostically eukaryotic feature, nor does the absence of preserved cellularity necessarily indicate a coenocytic condition. It's certainly a possibility, but would need to be accompanied by evidence of well-preserved cell walls more generally (e.g., in associated microfossils). In the absence of such criteria there are no grounds for assuming the individual 'beads' were not composites of multiple cells; if the constituent cells lacked secreted cell walls (like animals and most non-photosynthetic protists), then even that test would fail. Without some positive evidence for *Horodyskia* having a coenocytic construction, the extended discussion of various organisms that do or don't express such a condition carries no weight.

Response: As we mentioned above, even for the largest prokaryotes, sulfur bacteria *Thiomargarita namibiensis* and *Ca. Thiomargarita magnifica*, they are at least two orders of magnitude smaller than the largest beads ($2.3 \times 10^{10} \mu\text{m}^3$) of *Horodyskia* (Line 531-534). Only eukaryotic cells can reach the volume of the beads of *Horodyskia*. It is possible that the inner structure of the individual beads of *Horodyskia* are multicellular, but there is no evidence preserved, even for those three-dimensionally preserved organic-walled specimens (Fig. 21-p). On the contrary, based on observation of hundreds of beads, we are inclined to treat each bead of *Horodyskia* as a single cell, since paired beads (Fig. 5b, d, f) and possible binary fissions (Fig. 3b, e) are observed and consistent with such an interpretation.

320 – *Horodyskia* certainly could be a coenocytic alga, but it's simplistic to limit the search to extant groups with such a structure. It would make more sense to start by asking whether it was photosynthetic.

Response: It is very difficult to find direct evidence of nutrition modes for morphological simple Precambrian fossils, e.g., photosynthetic autotrophy (algae), heterotrophy (protozoans), or saprotrophy (some fungi). Aperture structures which allow pseudopodia to protrude through may be used as morphological evidence of a heterotrophic lifestyle for shelled protozoans such as testate amoebae and foraminifers. A recent study that has identified in situ bound nickel-geoporphyrins moieties as possible derivatives of chlorophyll offered a new approach to reveal fossil evidence for phototrophy (Sforna et al., 2022, *Nature Communications*, 13: 146). In our manuscript, we have tried to balance our discussion for possible affinity between heterotrophic protozoans (Line 548-577) and autotrophic algae (Line 578-595), because we did not have direct evidence for nutrition mode for *Horodyskia*, e.g., aperture structures or nickel-geoporphyrins moieties. We only treat coenocytic alga as one possible choice and try to give a parsimonious interpretation that *Horodyskia* represents a total-group multicellular eukaryote composed of a string of coenocytic cells (not necessarily photosynthetic) (Line 594-598).

337 – It would be useful to define/distinguish the terms multicellular and coenocytic at some point. Presumably it's the individual 'beads' of *Horodyskia* that are being interpreted as coenocytic, whereas the 'string' is multicellular?

Response: Yes, this understanding is what we meant. The sentence has been revised as "a parsimonious interpretation is that *Horodyskia* represents a total-group multicellular eukaryote that is composed of a string of coenocytic cells and achieved macroscopic size in the Proterozoic Eon" (Line 594-598).

362 – Yes, these various macroscopic taxa "may also be eukaryotes" – but the jury is still very much out. This should be more clearly acknowledged in the discussion (e.g., Sharma & Shukla 2009; Butterfield 2015; Porter 2020)

Response: The references about different views of these fossils have been cited in the revised manuscript (Line 621-623).

365 – there is nothing in these new Neoproterozoic examples of *Horodyskia* that adds weight to the argument for their eukaryoticity in the early Mesoproterozoic. At most, the presence of organic carbon supports the argument for biogenicity.

Response: This phrase has been removed as suggested by Reviewer #1 (Line 623-627). Because the early Mesoproterozoic materials are type materials and the new material herein are the same biological entity as the early Mesoproterozoic type materials. Reviewer #1 suggested that there is no reason we would exclude or not take into consideration the specimens from Montana (the type specimens).

Reviewer #3 (Remarks to the Author):

This manuscript presents well-preserved carbonaceous compressions or organic-walled microfossils of *Horodyskia* from the Tonian (~950- 720 Ma) Shiwangzhuang Formation in the western Shandong and Jiuliqiao Formation in the Huainan region of North China. Light microscopy (LM), Scanning electron microscope (SEM), Energy dispersive X-ray spectroscopy (EDS) elemental mapping, and Raman spectroscopy are used to ascertain the biogenicity, their taphonomic processes, and affinity to fossils.

The fossil material presented in the manuscript is novel and will be of interest to the Precambrian geoscience community because the *Horodyskia* is described as not more than half a dozen Proterozoic localities in the world.

The authors have done an excellent job describing this fossil material in detail and investigating the preservation of the organisms. All the illustrations in the manuscript are excellent. I think it could be accepted after minor revisions.

Comments:

Line no. 44: cite suitable reference.

Response: References have been added (Line 48-51).

Line no. 64: reference missing.

Response: References have been added (Line 70-73).

Reference no. 29, 35, 36, 37 and 53: incomplete (recheck).

Response: Corrected. The references in the list have been reformatted according to the standard Nature referencing style (Line 708-917).

Reviewers' comments:

Reviewer #1 (Remarks to the Author):

Comments on the Revised Manuscript, Tonian carbonaceous compressions shed light on Horodyskia, one of the oldest multicellular and coenocytic macro-organisms

Line-by-Line of New Draft:

Line 61: there is an error in the age of the Tonian cited here. If you want to continue using Ga, it should be 1 Ga–.72 Ga. As a matter of style: decimals for billions of years gets a bit odd. I think the reader would prefer to see "1000–720 Ma". It would be OK to mix *Ga* and *Ma* in the sentence-- the reader will follow it and "550–540 Ma" is more pleasant on the eyes than "0.55–0.54 Ga".

Line 140: the specimens of Yochelson and Fedonkin (2000) need to be listed in the synonymy- at least the type specimen of the type species

Lines 649-651: I think you might add a bit of hedging here here to avoid overstating things. "...these North China materials suggest that coenocytic protists *may have been* diverse in the Tonian Period when they became increasingly important in the paleoecology and geobiology during this time interval62,71." Similarly throughout- the idea of Horodyskia as a coenocyte is an interpretation-- not an established and undeniable fact-- and needs to be treated as such. Just a bit of softening language would suffice.

Line 758: You mention that you think Horodyskia shares similarities with both foraminifera and algae; these are two groups that are very distant from each other phylogenetically so you might need to point that out to the reader as you conclude.

Comments on Responses and changes made:

In general the authors have done a good job responding to my comments and those of the other reviewers. I appreciate the additional photos and data and I am glad to see the systematics section moved from the supplementary section into the main text.

One of my comments I feel still needs more attention asks the authors to spend time developing the reasoning for why the coenocytic body plan is the most parsimonious suggestion for Horodyskia, and how a coenocytic body plan matters. They do discuss modern coenocytes a bit, but what about Horodyskia indicates that it must be a coenocyte? In lines 551-552 the authors mention that the size of Horodyskia indicates it must be coenocytic because of its size-- Is its size enough?- just that great a volume of cytoplasm requires multiple nuclei? Can you develop this a bit more? At the end of that paragraph it seems like the field of modern analogs is then narrowed based on size- is it becoming circular in reasoning here? It might seem that I am picking on this issue too much, but it is really at the crux of your assertions for the whole paper.

I would also like to have the authors spend a sentence or two telling me what about its being coenocytic is significant? How does a coenocyte -or population of coenocytes- affect other organisms in the habitat or other attributes of the environment in a way that non-coenocytic protists cannot? In lines 619-620 the authors do mention carbon sequestration and that is along the lines of what I'm seeking, but needs to be more developed. Is there something special coenocytes do in the modern world?

Reviewer #2 (Remarks to the Author):

N/A

Reviewer #1 remarks about changes made for reviewer #2:

For the most part I find that the authors have addressed Reviewer #2's concerns. Below I list a small error I noticed and I make two suggestions- one is to add to a sentence in a way that makes it even more clear to the reader that the authors have considered points like those Reviewer #2 brings up, and the other is a suggestion of two papers the authors might consider when discussing divergences within the Amoebozoan testate amoebae. I don't see any of these suggestions as deal-breakers, but I think these small changes would enrich the paper.

--

a small error I noticed: In the author institutions, the State Key Laboratory is marked as 44, whereas it should be simply 4

in line 633 I would ask the authors to change this closing sentence slightly to reiterate all of the reasons they reject the prokaryotic affinity for Horodyskia, not just the cell-wall recalcitrance. Additionally, a point Reviewer 2 stressed is that the spectrum of sizes and cell-wall structures in extant prokaryotes may not encapsulate all of the variation possible over the past ~1.5 billion years. The authors could make a nod to acknowledge this by including a short clause in a sentence to that effect-- probably as a part of the paragraph-closing sentence in line 633. (There's no way to know all of the possible abilities of prokaryotes over this time period, but by acknowledging this to the reader, you're telling us you've thought about that too.) And in line 698 you could treat this as a call-back-- just as the spectrum of modern prokaryote morphologies may not illustrate the extent of their capabilities over the past ~1.5 billion years, so too the spectrum of modern coenocytes likely represents only a fraction of morphotypes from the past ~1.5 billion years.

With reference to the possible divergence times of testate amoebae, the authors might want to consider looking at Lahr et al (2019: Current Biology <https://doi.org/10.1016/j.cub.2019.01.078>) and Porter and Riedman (2019: Current Biology <https://doi.org/10.1016/j.cub.2019.02.003>) that I think make a more specific case about not just the origins of testate amoebae within the Amoebozoa, but looking in particular at clades within that larger group.

Reviewer comments in black fonts.

Point-to-point responses to reviewers' comments and quotes from revised manuscript are in red fonts, with line numbers (Line xxx) of the revised manuscript cited at the end of each response.

Reviewers' comments:

Reviewer #1 (Remarks to the Author):

Comments on the Revised Manuscript, Tonian carbonaceous compressions shed light on *Horodyskia*, one of the oldest multicellular and coenocytic macro-organisms

Line-by-Line of New Draft:

Line 61: there is an error in the age of the Tonian cited here. If you want to continue using Ga, it should be 1 Ga–.72 Ga. As a matter of style: decimals for billions of years gets a bit odd. I think the reader would prefer to see “1000–720 Ma”. It would be OK to mix *Ga* and *Ma* in the sentence-- the reader will follow it and “550–540 Ma” is more pleasant on the eyes than “0.55–0.54 Ga”.

Response: The age of the Backdoor and Stag Arrow formations of the Bangemall Supergroup in Western Australia is correct, which is late Mesoproterozoic (1.17–1.07 Ga), not Tonian (1000–720 Ma) (Cutten et al., 2021). We have changed the age “0.55–0.54 Ga” to “550–539 Ma” in this sentence and changes the age “~10–7.2 Ga” to “~1000–720 Ma) in the closing sentence as suggested (Line 58–63).

Reference:

Cutten, H., Zwingmann, H., Uysal, I. T., Todd, A. & Johnson, S. Report 214 Dating Proterozoic fault movement using K-Ar geochronology of illite separated from lithified fault gouge. (Geological Survey of Western Australia, 2021).

Line 140: the specimens of Yochelson and Fedonkin (2000) need to be listed in the synonymy- at least the type specimen of the type species

Response: “2000 *Horodyskia* Yochelson and Fedonkin⁶, p. 844.” has been added to the synonymy list of Genus *Horodyskia* as suggested (Line 136).

Lines 649-651: I think you might add a bit of hedging here here to avoid overstating things. “...these North China materials suggest that coenocytic protists *may have been* diverse in the Tonian Period when they became increasingly important in the paleoecology and geobiology during this time interval^{62,71}.” Similarly throughout- the idea of *Horodyskia* as a coenocyte is an interpretation-- not an established and undeniable fact-- and needs to be treated as such. Just a bit of softening language would suffice.

Response: Thanks for the suggestion. We have softened the language as suggested in several places (please see below).

Revised text:

- (1) “Nonetheless, it is also reasonable that we would not expect an evolutionary stasis experienced by eukaryotes for ca. 1.5 billion years and a parsimonious interpretation is that *Horodyskia* represents a total-group multicellular eukaryote that was composed of a string of giant-sized cells (probably coenocytic) and achieved a macroscopic size in the Proterozoic Eon.”(Line 612–616)
- (2) The Shiwangzhuang and Jiuliqiao materials indicate that *Horodyskia* acquired a large body size visible to the naked eyes probably through the combination of coenocytism and simple clonal coloniality. (Line 619–622)
- (3) “Considering that the earliest *Horodyskia* fossils are from the ca. 1.48 Ga Appekunny Formation in Montana²³, our study implies that macroscopic giant-celled (probably coenocytic) protists existed in the early Mesoproterozoic.”(Line 636–638)
- (4) “The interpretation of *Horodyskia* as a colonial giant-celled protist (probably coenocytic), considering its oldest occurrence dating back to ~1.48 Ga, adds to a short but growing list of early eukaryote fossils existed in this critical time interval, and suggests that a macroscopic coenocytic body plan has probably been present since the early Mesoproterozoic.” (Line 651–655)
- (5) “Together with the “string of beads” fossil *Horodyskia* herein and the siphonocladous *Proterocladus* from the early Tonian Nanfen Formation in North China⁶³, these North China materials suggest that giant-celled protists may have been diverse in the Tonian Period when they became increasingly important in the paleoecology and geobiology during this time interval^{73,83}.” (Line 662–666)
- (6) “Our study reports evidence that *Horodyskia*, a genus of macroscopic fossils ranging from the early Mesoproterozoic Era to the terminal Ediacaran Period, may have attained its macroscopic size through the combination of coenocytism and simple clonal coloniality.” (Line 679–682)

Line 758: You mention that you think *Horodyskia* shares similarities with both foraminifera and algae; these are two groups that are very distant from each other phylogenetically so you might need to point that out to the reader as you conclude.

Response: The phrase “although these two groups are phylogenetically distant and the latter are typically unicellular” has been added to the sentence here (Line 678–679).

Comments on Responses and changes made:

In general the authors have done a good job responding to my comments and those of the other reviewers. I appreciate the additional photos and data and I am glad to see the systematics section moved from the supplementary section into the main text.

One of my comments I feel still needs more attention asks the authors to spend time developing the reasoning for why the coenocytic body plan is the most parsimonious suggestion for *Horodyskia*, and how a coenocytic body plan matters. They do discuss modern coenocytes a bit, but what about *Horodyskia* indicates that it must be a coenocyte? In lines 551-552 the authors mention that the size of *Horodyskia* indicates it must be coenocytic because of its size-- Is its size enough?- just that great a volume of cytoplasm requires multiple nuclei? Can you develop this a bit more? At the end of that paragraph it seems like the field of modern analogs is then narrowed based on size- is it becoming circular in reasoning here? It might seem that I am picking on this issue too much, but it is really at the crux of your assertions for the whole paper.

Response: What matters here for the interpretation of *Horodyskia* is the large size ($\leq 2.3 \times 10^{10} \mu\text{m}^3$) and cellular activities (putative binary division and paired beads resulted from division) of its beads. It is these two important features that lead to the following conclusion that the beads of *Horodyskia* are at least two orders of magnitude larger than any extant and fossil prokaryotic cells and therefore are more likely large eukaryotic cells.

As far as we know, in extant protists (including foraminifera and algae), sub-millimeter- to millimeter-sized cells are usually multinucleated or coenocytic (e.g., Mine et al., 2008), although a coenocytic cell construction does not necessarily require a large cell size. Multiple nuclei for a giant cell is a consequence of the need for nuclear control of cellular activity (Briggs, 2022). Large size is difficult because in giant cells messages (mRNA, proteins) moving by diffusion take too long to get from the ‘control center’ (the nucleus, the ribosome) to all parts of the cell; therefore, multiple nuclei is required to solve this problem. However, we admit that technically large cell size does not equal to coenocyte, and we have made several changes and tried to discuss the possible affinity of *Horodyskia* in a more objective way by using “giant-celled” rather than “coenocytic” in the main text (e.g., Line 88; 596; 597; 600; 603; 604; 608; 609; 611; 615; 637; 652; 657; 664; 676).

The field of modern analogs is narrowed to simple colonial protists, because *Horodyskia* does not present morphological features that can be related to complex multicellular organisms such as brown algae, sponges and other metazoan, and fungi proposed in previous researches. Then we start to look for potential analogs in protists with large cell size. We do not think this is circular reasoning here. It is the construction of simple clonal coloniality, sub-millimetric to millimetric cell size (or inferred coenocyte), and recalcitrant cell wall of *Horodyskia* that together lead to the narrowed field of modern analogs. We have revised the discussion on modern analogs in a more detailed and logical way (Line 542–560).

References:

- Briggs, G. M. *Inanimate Life* (Second Edition). (Milne Open Textbooks, 2022)
- Mine, I., Menzel, D. & Okuda, K. Morphogenesis in giant-celled algae. *Int. Rev. Cell Mol. Biol.* 266, 37–83 (2008).

Revised text (Line 542–560):

“The large cell size of *Horodyskia* indicates that it is not only a eukaryote, but also likely a multinucleated or coenocytic eukaryote. The sub-millimeter- to millimeter-sized cells of *Horodyskia* seem to require multiple nuclei to regulate the giant mass of cytoplasm, because a single nucleus, via the diffusion of messenger RNAs, can only control a limited volume of cytoplasm^{1,40,41}. The cellular nature and the unconnected but occasionally dividing feature of *Horodyskia* beads, indicate that these fossils are unlikely to be complex multicellular organisms such as articulated brown alga¹⁵, branching sponge²², tissue-grade colonial metazoan²³, or endocyanotic fungus²⁴, but are more likely to be protists whose clonal cells forming simple and not fully integrated colonies (simple clonal coloniality; *sensu* ref. ¹). Multicellularity (including clonal and aggregative development) occurs in a number of eukaryotic groups, including fungi, animals, choanoflagellates, slime molds (dictyostelids, myxomycetes, protostelids, and acrasids), green algae, land plants, red algae, ciliates, oomycetes, diatoms, chrysophytes, xanthophytes, and brown algae⁴²⁻⁴⁴. However, protists with a construction of simple clonal coloniality, a sub-millimetric to millimetric cell size (or inferred coenocyte), and a recalcitrant cell wall, features that define *Horodyskia*, are limited to a smaller number of eukaryotic clades⁴⁵. Here we will evaluate the following three most likely potential analogs: arcellinid testate amoebae⁴⁶, foraminifers⁴⁵, and some algal groups⁴⁷⁻⁴⁹, which are proposed in previous studies and show some, if not all, of the features of *Horodyskia*.”

I would also like to have the authors spend a sentence or two telling me what about its being coenocytic is significant? How does a coenocyte -or population of coenocytes- affect other organisms in the habitat or other attributes of the environment in a way that non-coenocytic protists cannot? In lines 619-620 the authors do mention carbon sequestration and that is along the lines of what I’m seeking, but needs to be more developed. Is there something special coenocytes do in the modern world?

Response: In the revised manuscript, we have added the evolutionary importance of *Horodyskia* as a probable colonial and coenocytic protist (Line 619–628). Because it has combined coenocytism and simple clonal coloniality, and therefore stood at an evolutionary crossroad of the “unicellular-to-colonial-to-multicellular” transformation series and “coenocytic-to-multicellular” transformation series of body plans, as well as may represent a primitive condition for siphonocladous (multicellular multinucleate). It is not a coenocytic body plan, but a macroscopic body size that matters regarding to the environmental influence. A macroscopic body size, whether it is achieved by coenocyte, multicellularity, or the combination of both, can bring noticeable ecological advantages and enhance organic carbon burial and therefore facilitate marine ventilation. In the revised manuscript, we have added more details about the ecological advantages and environmental impact of macroscopic organisms as suggested (Line 629–634).

Revised text:

- (1) “The Shiwangzhuang and Jiuliqiao materials indicate that *Horodyskia* acquired a large body size visible to the naked eyes probably through the combination of coenocytism and simple clonal coloniality. A coenocytic body plan has been proposed as the direct progenitor of some multicellular algal, animal, and fungal groups, through subsequent process similar to segregative cell division, whereas a colonial body plan is traditionally regarded as progenitor of multicellular organisms⁶⁷⁻⁶⁹. It is interesting to note that *Horodyskia* may have stood at an evolutionary crossroad of the “coenocytic-to-multicellular” and “unicellular-to-colonial-to-multicellular” transformation series⁶⁹, and seems to represent a primitive condition for siphonocladous (multicellular multinucleate)⁶⁸.”(Line 619–628)
- (2) “Macroscopic body sizes can bring noticeable ecological advantages to eukaryotes, including increased speed and efficiency to occupy new adaptive niches or migration to more favorable environment, better protection against phagocytic predation, and increased possibility to capture larger preys⁷⁰⁻⁷²; larger eukaryotes can also act as a faster biological pump by accelerating the sinking flux and then enhancing the efficiency of organic carbon burial, and therefore facilitate marine ventilation^{71,73,74}.” (Line 629–634)

Reviewer #2 (Remarks to the Author):

N/A

Reviewer #1 remarks about changes made for reviewer #2:

For the most part I find that the authors have addressed Reviewer #2's concerns. Below I list a small error I noticed and I make two suggestions- one is to add to a sentence in a way that makes it even more clear to the reader that the authors have considered points like those Reviewer #2 brings up, and the other is a suggestion of two papers the authors might consider when discussing divergences within the Amoebozoan testate amoebae. I don't see any of these suggestions as deal-breakers, but I think these small changes would enrich the paper.

--

a small error I noticed: In the author institutions, the State Key Laboratory is marked as 44, whereas it should be simply 4

Response: I am afraid what the reviewer points to is the place with a marked change in the manuscript file of “Revised Manuscript with Changes Marked”. It is correct in our clean copy manuscript file where all the changes have been accepted (Line 13).

in line 633 I would ask the authors to change this closing sentence slightly to reiterate all of the reasons they reject the prokaryotic affinity for *Horodyskia*, not just the cell-wall recalcitrance. Additionally, a point Reviewer 2 stressed is that the spectrum of sizes and cell-wall structures in extant prokaryotes may not encapsulate all of the variation possible over the past ~1.5 billion years. The authors could make a nod to acknowledge this by including a short clause in a sentence to that effect-- probably as a part of the paragraph-closing sentence in line 633. (There's no way to know all of the possible abilities of prokaryotes over this time period, but by acknowledging this to the reader, you're telling us you've thought about that too.) And in line 698 you could treat this as a call-back-- just as the spectrum of modern prokaryote morphologies may not illustrate the extent of their capabilities over the past ~1.5 billion years, so too the spectrum of modern coenocytes likely represents only a fraction of morphotypes from the past ~1.5 billion years.

Response: The closing sentence has been revised as suggested (Line 538–541), to reiterate the reasons for rejecting a prokaryotic affinity for *Horodyskia* and to also acknowledge that we cannot rule out the possibility that some extinct prokaryotes could have developed giant-size cell and cell-wall recalcitrance.

Revised text (Line 538–541):

“Therefore, considering its extremely large cell size and inferred recalcitrant cell wall, *Horodyskia* is unlikely to be a prokaryotic organism, although we cannot entirely rule out the possibility that some extinct lineages of prokaryotes could have developed such a large cell size and cell-wall recalcitrance³⁹.”

With reference to the possible divergence times of testate amoebae, the authors might want to consider looking at Lahr et al (2019: Current Biology <https://doi.org/10.1016/j.cub.2019.01.078>) and Porter and Riedman (2019: Current Biology <https://doi.org/10.1016/j.cub.2019.02.003>) that I think make a more specific case about not just the origins of testate amoebae within the Amoebozoa, but looking in particular at clades within that larger group.

Response: These two references have been cited in the revised manuscript as suggested (Line 570–574).

REVIEWERS' COMMENTS:

Reviewer #1 (Remarks to the Author):

I think this paper is ready for publication. The authors have incorporated reviewer comments and considered and answered reviewer questions, developing a more nuanced look at Horodyskia in general, and these new specimens in particular.